# The Role of Molecular and Hormonal Factors in Obesity and the Effects of Physical Activity in Children

**DOI:** 10.3390/ijms232315413

**Published:** 2022-12-06

**Authors:** Jerónimo Aragón-Vela, Jesús Alcalá-Bejarano Carrillo, Aurora Moreno-Racero, Julio Plaza-Diaz

**Affiliations:** 1Department of Health Sciences, Area of Physiology, Building B3, Campus s/n “Las Lagunillas”, University of Jaén, 23071 Jaén, Spain; 2Department of Health, University of the Valley of Mexico, Robles 600, Tecnologico I, San Luis Potosí 78220, Mexico; 3Research and Advances in Molecular and Cellular Immunology, Center of Biomedical Research, University of Granada, Avda, del Conocimiento s/n, 18016 Armilla, Spain; 4Department of Biochemistry and Molecular Biology II, School of Pharmacy, University of Granada, 18071 Granada, Spain; 5Children’s Hospital of Eastern Ontario Research Institute, Ottawa, ON K1H 8L1, Canada; 6Instituto de Investigación Biosanitaria IBS, Granada, Complejo Hospitalario Universitario de Granada, 18014 Granada, Spain

**Keywords:** obesity, physical activity, hormones, children, adipose tissue

## Abstract

Obesity and overweight are defined as abnormal fat accumulations. Adipose tissue consists of more than merely adipocytes; each adipocyte is closely coupled with the extracellular matrix. Adipose tissue stores excess energy through expansion. Obesity is caused by the abnormal expansion of adipose tissue as a result of adipocyte hypertrophy and hyperplasia. The process of obesity is controlled by several molecules, such as integrins, kindlins, or matrix metalloproteinases. In children with obesity, metabolomics studies have provided insight into the existence of unique metabolic profiles. As a result of low-grade inflammation in the system, abnormalities were observed in several metabolites associated with lipid, carbohydrate, and amino acid pathways. In addition, obesity and related hormones, such as leptin, play an instrumental role in regulating food intake and contributing to childhood obesity. The World Health Organization states that physical activity benefits the heart, the body, and the mind. Several noncommunicable diseases, such as cardiovascular disease, cancer, and diabetes, can be prevented and managed through physical activity. In this work, we reviewed pediatric studies that examined the molecular and hormonal control of obesity and the influence of physical activity on children with obesity or overweight. The purpose of this review was to examine some orchestrators involved in this disease and how they are related to pediatric populations. A larger number of randomized clinical trials with larger sample sizes and long-term studies could lead to the discovery of new key molecules as well as the detection of significant factors in the coming years. In order to improve the health of the pediatric population, omics analyses and machine learning techniques can be combined in order to improve treatment decisions.

## 1. Introduction

In the context of health, overweight and obesity are defined as abnormal or excessive fat accumulations [1]. Overweight is defined as having a body mass index (BMI) over 25, while obesity is defined as having a BMI over 30 [2]. A global burden of disease study indicates that over 4 million people die each year as a result of being overweight or obese, a problem that has reached epidemic proportions [3]. In both adults and children, overweight and obesity are on the rise. Overweight or obesity increased four-fold, from 4% to 18%, among children and adolescents aged 5–19 years between 1975 and 2016 [4].

It is possible to calculate BMI in adults to determine obesity; however, it is more difficult in children to determine it accurately [5]. Overweight and childhood obesity are currently classified according to three classification systems. They include the criteria published in 2007 by the World Health Organization (WHO) [6], the data tables published by the Centers for Disease Control and Prevention (CDC) in 2000 [7], and the guidelines published by the International Obesity Task Force (IOTF or Cole-IOTF) [8]. According to the WHO, overweight is considered as a body weight greater than one standard deviation above the median of the WHO reference growth standard. Generally, obesity is classified as a weight over two standard deviations above the median of the WHO growth reference standard for height [6]. Overweight is defined by the CDC as a BMI exceeding the 85th percentile but below the 95th percentile. Obesity is defined as a BMI exceeding the 95th percentile, according to age and gender [7]. Using the cut-offs specified for adults as a basis, Cole et al. developed BMI curves identifying overweight as 25–30 kg/m^2^ and obesity as >30 kg/m^2^ by age group and gender [8]. Because of its simplicity and low cost, BMI has become an increasingly popular indicator for identifying overweight and obesity [9]. BMI does not distinguish among increased mass in the form of fat, lean tissue, or bone, and, consequently, it may lead to significant misclassification, especially in children and adolescents [10]. Consequently, other electrical bioimpedance measurements are required in order to provide a more complete estimate of additional body compartments [11,12].

In every region except sub-Saharan Africa and Asia, obesity is the primary cause of malnutrition today. In low- and middle-income countries, particularly in urban settings, overweight and obesity are rapidly increasing [13]. Once considered a problem only in high-income countries, overweight and obesity have become a major public health concern. Children living in developing countries are more likely to be overweight or obese, with a rate of increase higher than that of developed countries [14].

There are a number of preventable and reversible causes of overweight and obesity. The growth of this epidemic has, however, not yet been reversed in any country [15]. In spite of the presence of other factors, obesity is fundamentally caused by an imbalance between calories consumed and calories expended. Over the past few decades, there has been an increase in the consumption of high-fat, high-sugar foods with high energy density [16]. Increasing urbanization and the changing nature of many types of work have also contributed to a decline in physical activity [17].

Increasing the portion of daily intake of fruit, vegetables, legumes, whole grains, and nuts is one way to lower the risk of overweight and obesity. It is also important to engage in regular physical activity (60 min per day for children and 150 min per week for adults) [18]. Infants who are exclusively breastfed from birth to six months of age are less likely to become overweight or obese, according to studies [19].

Adipose tissue is more than simply a collection of adipocytes; each adipocyte is closely connected to the extracellular matrix (ECM) [20,21]. Among the noncellular components of the ECM are collagens, laminins, fibronectins, nidogens, and proteoglycans. There are many types of these components and their structures are complex. In addition to providing a unique and critical microenvironment for adipocytes, ECM molecules regulate the bioactivity of these cells [20,21,22].

Excess energy is stored in adipose tissue through expansion. Adipocyte hypertrophy and hyperplasia lead to obesity, which results from the abnormal expansion of adipose tissue [22]. Besides causing adipose tissue dysfunction and ectopic lipid accumulation, unhealthy expansion can also cause inflammation and metabolic diseases, such as diabetes type 2 and insulin resistance [22,23]. Healthy expansion of adipose tissue depends on the reorganization of the ECM, whereas fibrosis is a key feature of inflammation and dysfunction in adipose tissue [24]. A significant amount of research is still needed to assess the direct effects of impaired cell–matrix interaction on adipocyte function and insulin sensitivity [24].

Adipocytes and adipose tissue are stimulated to accumulate and perform adipocyte functions by estrogens, either directly or indirectly. According to evidence, estrogen augments the sympathetic tone differentially according to the adipose tissue depot. This favors lipid accumulation in the subcutaneous depot in women and visceral fat deposition in men. In the context of adipocyte function, estrogens and their receptors are known to affect the extent to which fat cells expand. They achieve this by enhancing it at the level of the subcutaneous depot and inhibiting it at the level of the abdominal depot [25].

There are several hormones and neurotransmitters that are involved in regulating appetite and energy expenditure, such as leptin, cocaine- and amphetamine-regulated transcription, and ghrelin. There are specific brain centers that are affected by these hormones that are responsible for controlling the sensation of satiety [26]. There is a possibility that weight gain can be triggered by mutations in these hormones or their receptors [27]. It has been estimated that 40–70% of the genetic risk of obesity is associated with this disorder. Several genetic loci have been found to be related to BMI and obesity risk as a result of genome-wide association studies [28].

Currently, the American College of Sports Medicine (ACSM) recommends either aerobic or anaerobic exercise. The term “aerobic exercise” refers to exercises that exhaust the muscles’ oxygen supply [29]. However, oxygen consumption is sufficient to supply the muscles with the energy that they need, without the need for any other source of energy [30]. Alternatively, in anaerobic exercise (weight lifting or resistance training), oxygen consumption is inadequate to meet the energy demands placed on the muscles, and the muscles are forced to break down other energy sources, such as sugars, in order to generate energy and lactic acid [31]. In addition to physical activity, structured exercise plans and sessions may also be included in the exercise program [32].

Following COVID-19, the incidence of obesity in children has increased worldwide, and the WHO has noted the various consequences of this problem [33]. As a result of the current pandemic, children have been advised to remain at home. This results in restricted mobility and a more sedentary lifestyle, which contributes to weight gain and diminishes the overall function of the body [34]. Children and adolescents, who are at the beginning of their physical, intellectual, and emotional development, have been particularly affected by the COVID-19 pandemic [35,36]. In Europe, approximately 20% of children aged 2–19 years are overweight or obese, which has negative health consequences in adulthood [37].

A comprehensive literature search was conducted using electronic databases, including Medline (PubMed), EMBASE, and the Cochrane Library. The keywords that we used were “obesity”, “physical activity”, “exercise”, “hormones”, “childhood obesity”, “children”, “extracellular matrix”, and “adipose tissue”. For additional relevant literature, we searched the reference lists of the articles included in the review.

Therefore, the present review aimed to investigate the molecular and hormonal mechanisms recently analyzed in obesity. In addition, the present review aimed to investigate the involvement of leptin in physical exercise and its relevance in the control of obesity, especially in the child population.

## 2. Molecular Control in Obesity

Adipose tissue in mammals can be divided into two types: white adipose tissue (WAT), which stores energy, and brown adipose tissue (BAT), which generates heat [38,39,40]. In adults, BAT has limited therapeutic potential due to its capacity for energy expenditure. However, the significant capacity of BAT for energy expenditure is an interesting mechanism for the treatment of metabolic diseases [41]. The metabolic alterations associated with obesity have not been studied separately in the context of adipose tissue using global omics methods. Transcriptomics studies have shown that obesity is associated with the significant dysregulation of adipose tissue [42]. Mitochondrion-related pathways are downregulated in the adipose tissue transcriptome.

A variety of pathways are involved, including oxidative phosphorylation (OXPHOS), the catabolism of branched-chain amino acids (BCAAs), fatty acid β-oxidation [43,44,45], and the upregulation of inflammatory [43,44,45,46,47] and ECM organization pathways [46,47]. In individuals with more severe insulin resistance, these findings become more obvious [45,47,48,49]. 

As well as these findings, microarray analyses of mature adipocytes [50,51,52] and metabolome analyses of adipose stem cell cultures [53] have revealed that obesity is associated with changes in glucose and amino acid metabolism, mitochondrial metabolism, and inflammation. Factors such as the environment and genetics influence the storage of adipose tissue. An individual’s lifestyle, such as his or her food intake and physical activity, is generally influenced by the environment. Twin studies, adoption studies, and segregation analyses have demonstrated the importance of genetic factors in obesity [42,54,55,56].

### 2.1. Structure and Components of ECM Related to Obesity

Bidirectional signaling between the cell adhesion to ECM proteins and cell–cell adhesion is mediated by the integrin families of cell surface receptors. A number of studies have demonstrated interactions between integrin receptors and other growth factor receptor tyrosine kinases [57,58,59,60] that modulate downstream signaling [61,62,63]. They have been extensively studied since their discovery by Hynes in 1987 [64], and they transduce signals through the plasma membrane for the activation of intracellular signaling. A transmembrane receptor called an integrin consists of both heterodimeric subunits, which are composed of α- and β-subunits [65]. There are 24 distinct types of integrins, each of which has different binding specificities and signaling properties [8]. An integrin contains a large ectodomain that facilitates interactions with ligands, a transmembrane domain, and a cytoplasmic tail that binds to the actomyosin cytoskeleton indirectly [20].

In order to activate them, a change from a bent-closed to extended-closed conformation is necessary [20]. The activation of integrins is dependent upon intracellular adaptor proteins, such as talins and kindlins. Integrins themselves do not have kinase activity, and downstream signaling occurs through focal adhesion kinases (FAKs) and integrin-linked kinases (ILKs) [66,67]. Tyrosine kinases such as FAK are involved in intracellular signaling, cytoskeleton stabilization, and focal adhesion turnover. They are regulated by epidermal growth factor receptors (EGFRs), fibroblast growth factor receptors (FGFRs), and insulin receptors [20].

Adipose tissue FAK signaling is believed to control insulin sensitivity by regulating adipocyte survival [65], whereas ILK interacts with the cytoplasmic domains of integrins, as well as numerous cytoskeleton-associated proteins. A wide variety of proteins and enzymes bind to integrin receptors, such as collagen, fibronectin, laminin, Arg–Gly–Asp peptides (RGD), and leucocytes [68].

Approximately 50% of the non-cell mass of adipose tissue is composed of collagen, which is the main component of the ECM. The adipocytes are primarily responsible for the production of collagen, although endothelial cells, preadipocytes, and stem cells can also produce it [69,70]. The strong external skeleton of mature adipocytes can reduce the mechanical stress generated by storing energy as triglycerides. This is transferred from the outside to the inside of the cell. As well as contributing to cell adhesion, migration, differentiation, morphogenesis, and wound healing, collagen is an extremely crucial structural component of adipose tissue. In the basement membranes of adipocytes, collagen IV plays a major role in their survival [69,70].

Matrix metalloproteinases (MMP), metalloproteinase domain-containing protein(ADAM), and ADAM with a Thrombospondin type-1 motif are subfamilies of the metzincin superfamily of zinc-dependent metalloproteinases [71]. Among the most critical functions of MMPs is the degradation of ECM proteins using calcium-dependent and zinc-containing endopeptidases [72,73]. As part of wound healing, angiogenesis, and tumor cell metastasis, MMPs play a key role in regulating ECM remodeling in both normal physiology and disease [73,74]. As well as expanding adipose tissue, causing liver fibrosis, and promoting atherosclerosis, MMPs also play an integral role in other biological processes [75].

Specific studies in children have shown that the plasma levels of MMP-9 and TIMP metallopeptidase inhibitor 1 (TIMP-1) are elevated in obese children and adolescents [76]. Another study evaluating MMPs found that plasma MMP9 concentrations were significantly higher in preterm babies when compared to term babies. Preterm deliveries were associated with almost three-times-higher plasma MMP9 levels when compared to controls [77]. In children with obesity, MMP-9 genotypes and haplotypes are associated with higher levels of MMP-9, suggesting that genetic factors can alter the relevant pathogenic mechanisms involved in the development of cardiovascular complications associated with obesity in childhood [78]. For MMP-2 polymorphism, similar results have been observed, with CC genotypes being more common in subjects (controls and children with obesity) with higher MMP-2 concentrations, while CT genotypes and the T allele are less common in children with obesity [79]. In addition, the expression of genes related to the regulation of the extracellular matrix (*TNMD* and *NQO1*) in visceral adipose tissue was higher in children with obesity when compared with normal-weight prepubertal children [80].

Currently, the information about ECM proteins and obesity comes from in vitro study models, and there is a need for more research that utilizes large populations of children.

### 2.2. Control of Differentiation in WAT

In terms of metabolism, WAT primarily stores triglycerides, which can be released during times of energy demand, such as starvation or exercise [81].

The differentiation of mesenchymal stem cells into mature WAT takes place under the control of various cellular receptors that act as transcription factors (Peroxisome proliferator-activated receptor gamma, PPARγ; Peroxisome proliferator-activated receptor alpha, PPARα; Peroxisome proliferator-activated receptor beta/delta, PPARβ/δ; Retinoid X receptor (heterodimer with PPARγ) RXR [82]) and play a role in four physiological processes: proliferation, mitotic cloning, early differentiation, and terminal differentiation. 

Other key molecules that are involved in differentiation are CCAAT/enhancer-binding protein (C/EBP) family members (i.e., C/EBPα, C/EBPβ, and C/EBPδ), Krox20, KLFs, CREB, and C/EBPβ and δ; they act early in terminal differentiation to induce the expression of PPARγ and C/EBPα [83]. PPARγ and C/EBPα are responsible for providing a positive feedback mechanism that locks differentiation in place [84,85]. Another form of positive feedback is produced by PPARγ and C/EBPβ [86].

The liver-based receptors direct the deposition of energy into adipose tissue for storage through alterations in liver metabolism [87]. Additionally, systemic hormones that bind to their receptors in the liver, brain, and adipose tissue are able to alter food intake versus energy intake (insulin receptor, estrogen receptors (α, β), androgen receptor, glucocorticoid receptor, and thyroid hormone receptors (α, β)) [82].

Signal cascades involved in promoting the osteogenic or adipogenic differentiation of mesenchymal stem cells are typically mediated by one or both of the two transcription factors RUNX2 and PPARγ. PPARγ is generally regarded as the master regulator of adipogenesis, with an anti-osteoblastogenic effect that is also well documented [88]. The master regulator of osteogenesis, on the other hand, is known as RUNX2 [89]. Alkaline phosphatase (ALP) and osteocalcin (OCN) are activated by RUNX2 and contribute to collagen type I synthesis [90]. Several studies have shown that RUNX2 inhibits adipogenesis when overexpressed [91].

The expression of pluripotent genes, adipogenic transcriptional factors, RUNX2, inflammatory mediators, insulin signaling proteins, and adipogenic and osteogenic differentiation was compared in subjects with obesity and in healthy controls [92]. The cells derived from the human adipose tissue of individuals with obesity showed diminished proliferation, altered pluripotent genes, increased inflammation, and decreased osteogenesis [92].

As in the case of studies related to ECM and obesity, few studies have included populations of children. Currently, a variety of novel proteins are being discovered and linked to obesity [93,94,95], as childhood is often the optimal timeframe for treating obesity.

### 2.3. Genetics Underlying Childhood Obesity

The prevalence of childhood and adolescent obesity has reached epidemic proportions in the United States [96]. Approximately 17% of US children are obese at present. There are a number of aspects of a child’s health that can be affected by obesity, including psychological and cardiovascular health; in addition, their overall physical health will also be affected by obesity. Obesity is a public health concern for children and adolescents due to its association with other health conditions [96]. Ahmad et al. state that eight out of ten adolescents aged 10 to 14 years, twenty-five percent of children aged under 5, and fifty percent of children aged 6 to 9 years are at risk of becoming obese adults [97]. Despite the fact that genetic influences likely operate within a wide range of weight groups, they are particularly dominant during childhood-onset obesity and at either end of the BMI distribution, i.e., underweight or severe obesity [98].

Metabolomics studies have provided insights into the existence of unique metabolic profiles in children who have obesity [99,100]. It was observed that there were abnormalities in several metabolites related to lipids, carbohydrates, and amino acids as a result of low-grade inflammation in the system [101]. The carbon metabolism in obese children shifts toward hypoxic conditions. 

High levels of pyruvate, lactate, and alanine were detected in resting blood samples, indicating altered metabolism in the early years of a child’s life [102,103,104]. Elevated pyruvate levels suggest a deficiency in the enzyme pyruvate dehydrogenase (complex), which is needed to make acetyl-CoA going forward [101]. Regarding lactate increases, it could show deregulations in central carbon metabolism and a tendency to direct metabolism towards fermentation conditions in children with obesity, which is called “aerobic glycolysis” or the “Warburg effect” [105]. Therefore, from a physiological point of view, obesity corresponds hand in hand with adipocyte hypertrophy, which is associated with local hypoxia, which enhances lactate production [106]. As a result of the heterogeneities observed between the studies reviewed, ethnicity, diet, and physical activity were identified as significant influencing factors regarding the metabolome and lipidome [101]. Obesity and obesity-related complications are linked to visceral adipose tissue. Despite this, it is unclear how visceral adipose tissue interacts with obesity-related metabolic complications [107]. 

Studies with a cross-sectional observational design strongly suggest that childhood obesity or childhood leanness is associated with certain family lines. The weight status of parents is strongly correlated with the weight status of their offspring up to the age of five [108]. In addition, it is associated with the risk of obesity in adulthood. 

There is an association between birth weight and the risk of obesity later in life, with low- and high-birth-weight babies being at greatest risk [108]. In addition, several genetic mechanisms are involved in the life-course association between early growth phenotypes and adult cardiometabolic disease [109].

Studies of twins, families, and adoptions have estimated the heritability of obesity to range between 40% and 70% [110,111,112]. In line with this fact, genetic approaches can be utilized in order to characterize the physiological and molecular mechanisms that are responsible for controlling weight [110,111,112,113]. It is important to note that the evidence for genes related to childhood obesity is very broad and may differ based on age range and gender [114,115]. For this reason, Table 1 summarizes the main variables of the included studies that dealt with genetics and the onset of childhood obesity.

The authors of a meta-analysis published in 2010 found that genetic factors play an influential role in the variation in BMI at all ages. A substantial effect was also seen in middle childhood by common environmental factors, but this effect disappeared in adolescence [116]. *FTO* was the most important gene involved in obesity, which was also implicated in the regulation of adipocyte thermogenesis [117].

Methylation of the long non-coding RNA ANRIL (encoded at *CDKN2A*) was associated with adiposity in the birth tissue of ethnically diverse neonates, the peripheral blood of adolescents, and the adipose tissue of adults. Perinatal methylation at gene function loci may serve as a robust indicator of later adiposity [118].

A total of ten polymorphisms were evaluated in 730 Portuguese children aged 6 to 12 in order to assess their vulnerability to obesity. Methionine sulfoxide reductase A (*MSRA*), transcription factor AP-2 beta (*TFAP2B*), melanocortin 4 receptor (*MC4R*), neurexin 3 (*NRXN3*), peroxisome proliferator-activated receptor gamma coactivator 1 alpha (*PPARGC1A*), transmembrane protein 18 (*TMEM18*), homolog of Sec16 (*SEC16B*), homeobox B5 (*HOXB5*), and olfactomedin 4 (*OLFM4*) were evaluated [119]. The results of this study suggest that polymorphisms of the *MC4R, PPARGC1A, MSRA*, and *TFAP2B* genes may be associated with obesity-related traits in a sample of Portuguese children [119].

Based on a meta-analysis that included 12 eligible studies with 5000 cases and 9853 controls, the FTO rs9939609 polymorphism was significantly associated with an increased risk of obesity [120].

According to the meta-analysis of ALSPAC and Raine samples, a novel single-nucleotide polymorphisms (SNP) was detected downstream of the *FAM120AOS* gene on chromosome 9 [121]. The association was triggered by differences in BMI at 8 years (T allele of rs944990 increased BMI, with a modest association with change in BMI over time) [121]. A locus associated with childhood obesity (*OLFM4*) has reached genome-wide significance in relation to BMI at age 8 and/or changes over time [121].

In addition, there is a strong association between the exonic rs8192678-T SNP of *PPARGC1A* and a reduction in BMI z-score [122]. Another study has shown that *SEC16B* and *TMEM18* were associated with 27% and 40% increased odds of obesity, respectively, in Hispanic/Latino children (22–88% frequency) [123]. According to the IDEFICS/I.Family study, significant associations were found for five SNPs for the *FTO* and *CETP* genes [124].

Several studies have shown a significant association between the *FTO* rs9930506 and *MC4R* rs17782313 polymorphisms and obesity in children [125]. These genetic variants were associated with childhood obesity in Caucasians and Asians, according to stratified analyses [125]. According to a recent systematic review, the polymorphisms rs9939609 *FTO* and rs17782313 *MC4R* could be associated with overweight and obesity in children and adolescents. This depends on the study population and ethnicity [126].

**Table 1 ijms-23-15413-t001:** Genetics and the onset of childhood obesity.

Gen or Polymorphism	Type of Polymorphism	Type of Study	Age	Reference
In a UK twin sample, the *FTO* gene was found to be associated with BMI beginning at age seven, and a similar finding was also noted in Danish children. In the UK study, the *FTO* gene explained only a small portion of the genetic variation in BMI.	-	Systematic review, nine twin and five adoption studies	up to 18 years old	[116]
ANRIL methylation (encoded by *CDKN2A*) has been associated with adiposity in the birth tissue of ethnically diverse neonates, peripheral blood of adolescents, and adipose tissue of adults.	-	Prospective study, randomized control trial	From birth to 6 years old	[118]
It may be assumed that *MC4R, PPARGC1A, MSRA,* and *TFAP2B* genes contribute to obesity risk in this sample of Portuguese children.	-	Prospective study with 730 children	2 to 6 years old	[119]
*FTO* rs9939609 polymorphism was significantly associated with an increased risk of obesity.	SNP	Meta-analysis with 12 eligible studies with 5000 cases and 9853 controls	up to 18 years old	[120]
*FAM120AOS* gene was triggered by differences in BMI at 8 years (T allele of rs944990 increased BMI, with a modest association with change in BMI over time). A locus associated with childhood obesity (*OLFM4*) has reached genome-wide significance in relation to BMI at age 8 and/or changes over time.	SNP	GWAS meta-analysis of BMI trajectories from 1 to 17 years of age in 9377 children	1 to 17 years of age	[121]
A strong association between the exonic rs8192678-T SNP of *PPARGC1A* and a reduction in BMI z-score.	SNP	Randomized, prospective, double-blind, placebo-controlled, multicenter trial, 134 children	aged 7 to 14	[122]
*TMEM18* and *SEC16B* were associated with an increased risk of obesity of 27% and 40%, respectively, in Hispanic/Latino children (22–88% frequency).	SNP	Genome-wide association study of childhood obesity in 1612 Hispanic/Latino children and adolescents	2 to 18 years of age	[123]
Polymorphisms at *FTO* rs8050136 and *CETP* rs708272 have been identified as significant with childhood metabolic syndrome.	SNP	Prospective cohort study with 3067 children	aged 2 to 10 years	[124]
*FTO* rs9930506 and *MC4R* rs17782313 polymorphisms and obesity in children.	SNP	Meta-analysis with 13 studies on *MC4R* rs17782313 and 18 studies on *FTO* rs9939609	up to 18 years old	[125]
*FTO* rs9939609 and *MC4R* rs17782313 polymorphisms have been associated with overweight and obesity in children.	SNP	Meta-analysis with 12 studies on *MC4R* rs17782313 and *FTO* rs9939609	up to 18 years old	[126]

Abbreviations. BMI, body mass index; SNP, single-nucleotide polymorphisms.

A number of novel analyses have been introduced to identify functional enrichment pathways, protein–protein interactions, and network analyses that aim to identify pathways that target transcription factors, microRNA, and regulatory networks [107]. An overview of these studies is provided below.

Using public datasets, one hundred and eighty-four differentially expressed genes in the visceral adipose tissue were identified in children with obesity based on the selected datasets [107]. From these 19 candidate hub genes, 19 target transcription factors and miRNAs were analyzed. Several miRNAs identified in this study are linked to obesity-related pathways and diseases [107].

According to the DisGeNET search, 191 genes are associated with childhood obesity [127,128]. The protein–protein interaction network resulting from the study contained twelve hub-bottleneck genes (*INS*, *LEP*, *STAT3*, *POMC*, *ALB*, *TNF*, *BDNF*, *CAT*, *GCG*, *PPARG*, *VEGFA*, and *ADIPOQ*), as well as four functional clusters, with cluster 1 exhibiting the highest interaction score [127,128]. There was an enrichment of genes associated with inflammation, carbohydrate metabolism, and lipid metabolism within this cluster. With the exception of *POMC*, all hub-bottleneck genes were found in cluster 1, which contains genes with a high degree of connectivity that may play key roles in obesity-related pathways [127,128].

After adjusting for sex and maturational status, 256 genes were differentially expressed between fit and unfit children with overweight/obesity [129]. By analyzing the enriched pathways, gene pathways associated with inflammation were identified (e.g., the dopaminergic and GABAergic synapse pathways). A set of differentially expressed genes was identified by in silico validation data mining to be associated with cardiovascular disease, metabolic syndrome, hypertension, inflammation, and asthma [129].

Approximately 2% to 5% of individuals with childhood obesity have heterozygous mutations in the *MC4R* gene, making it the most common gene for which highly penetrant variants contribute to obesity [130,131].

The results of a follow-up study involving 1082 participants aged 12–13 years showed that skinfold thickness was associated with gene variants being lower in children with the *MC4R* TT genotype and the *LEP* AG genotype, but no associations were found with BMI [132].

Certainly, there have been studies that have demonstrated that the genes associated with severe obesity in experimental animals are also involved in childhood-onset obesity in humans [133], often without developmental delay [98]. These mutations and their carriers have been functionally and physiologically characterized in a manner that has demonstrated a high degree of convergence between mechanisms that regulate energy balance among mammalian species [98]. However, there is a need for more studies with larger sample sizes in order to fully understand how genetics contribute to childhood obesity.

## 3. Hormonal Control in Obesity

As a result of lipolysis, white adipocytes in WAT release a substantial amount of free fatty acids (FFAs), causing elevated levels of FFA in the serum of obese subjects. It has been hypothesized for several decades that this overflow of lipids from obese adipose depots contributes to obesity-associated insulin resistance and hepatosteatosis [67,68]. Although FFA are often considered as a whole in this context, studies exploring the impacts of individual lipid species have provided intriguing insights into adipocyte-secreted lipid specificities [69]. The discovery of leptin as an adipokine secreted by adipose tissue (adipokine) with potent anorexic properties in 1994 redefined WAT as an endocrine organ [70]. Adipokines have been identified as critical regulators of lipid and glucose homeostasis over the past two decades, and this list continues to grow. Rosen and Spiegelman (2006) [71] suggest that adipokines play an instrumental role in mediating the interaction of adipose tissue with other key metabolic organs, including the liver, muscle, and pancreas. Adipokine pathway dysfunction often leads to impaired organ communication and metabolic abnormalities in multiple tissues, which constitute a critical pathology contributing to metabolic disease [72].

### 3.1. Leptin

Appetite and related hormones, such as leptin, play an active role in food intake regulation and in the occurrence of obesity in children. Indeed, leptin plays a key permissive role in the onset of puberty in boys and girls, given that leptin injection can significantly accelerate the onset of puberty in juvenile female mice, by directly interacting with the KiSS-1gene [134]. Thus, given the importance of the timing of normal puberty in boys and girls, normal body weight and composition must be attained during childhood to avoid pubertal dysfunction. 

Leptin is a 16 kDa cytokine that acts as a satiety signal in the central nervous system and is related to glucose and insulin metabolism [34]. It has been suggested that it could be the link between obesity, diabetes, and cardiovascular risk [34]. The hypothalamic leptin–melanocortin system is a key element in the regulation of hunger, satiety, and energy balance [35]. Thus, genetic mutations that cause the absence of leptin or the leptin receptor (LepR) produce metabolic changes that resemble those observed during an intense negative energy balance, including hunger and the suppression of physiological processes that expend energy [135]. Furthermore, genetic mutations could also alter the dynamics of the release of FSH and LH secretion from the pituitary gland [136].

Leptin, mainly synthesized in adipocytes, exerts multiple functions. One key function is as one of the key modulators in regulating food intake and energy metabolism, by provoking energy expenditure and suppressing food intake via the hypothalamus [36]. During weight maintenance, the most significant determinant of circulating leptin concentrations is body fat mass and, indirectly, BMI [37]. Furthermore, leptin was shown to regulate the production of neurohormones in the medio-basal hypothalamus, such as thyrotropin-releasing hormone (TRH) neurons of the periventricular nucleus [137]. An increase in TRH release was shown to lead to the higher pituitary secretion of TSH, which, in turn, stimulates thyroid function and proliferation [138]. Moreover, leptin receptors have been found in the anterior pituitary and thyroid gland. Direct inhibitory actions on TSH secretion and on the expression of the Na^+^/I^–^ symporter and thyroglobulin mRNA in thyroid cell lines have been reported [139]. 

On the other hand, leptin levels fall rapidly in response to fasting and evoke profound changes in energy balance and hormone levels. Low leptin levels induce overfeeding and suppress energy expenditure, thyroid and reproductive hormones, and immunity [140]. In addition, leptin may also be critical for glycemic control, independent of its effects on calorie intake and energy expenditure [141]. Indeed, there is growing evidence from animal models that leptin plays a critical role in type 2 diabetes prevention and control by promoting beta-cell function and survival, improving insulin sensitivity, and regulating glucose metabolism [142]. This is because leptin is involved in the regulation of glucose homeostasis and insulin sensitivity. This is because these two hormones exert opposing effects and regulate each other, so that leptin inhibits insulin, and insulin stimulates leptin synthesis and secretion [143].

Humans with impaired leptin signaling exhibit intense hunger (hyperphagia), reduced sympathetic tone, mild hypothyroidism, hypogonadism, and impaired immune system functions. Those who have mutations in the leptin gene are treated with recombinant human leptin administration [144,145,146,147]. In obese mice and humans with disrupted leptin signaling, blood pressure is low compared to diet-induced obesity in rodents and obesity in humans [148], and leptin appears to play a major role in the increase in blood pressure that occurs with weight gain [98].

Patients with genetic mutations in the leptin gene or leptin receptor were found to be obese, and the chronic repositioning of leptin caused the normalization of their body weight [138]. However, most obese patients have hyperleptinemia but are resistant to the anorexigenic central action of leptin [138]. Nowadays, the precise mechanism through which hyperleptinemia leads to leptin resistance is not known. The causes of leptin resistance appear to be heterogeneous but also include constitutive defects in the neural circuit downstream of leptin [149]. Indeed, to date, all genes identified as Mendelian causes of obesity in humans are expressed in the central nervous system. Most of these genes are components of the neural circuit modulated by leptin [149].

### 3.2. Adiponectin

Adiponectin has a 30 kDa molecular weight and is secreted exclusively by WAT adipocytes [150], and abundant levels in the circulation are eliminated by the liver. It is significant to note that adiponectin is more abundant in subcutaneous WAT than in visceral WAT, which makes it the most abundant and adipose-specific adipokine [151]. Adiponectin, however, decreases before obesity and insulin resistance develop, and it attenuates inflammation and insulin resistance [152]. Adiponectin is thought to play a significant role in the pathogenesis of obesity, insulin resistance, and insulin resistance [152]. Furthermore, adiponectin is a hormone with cardioprotective functions [153]. The cardioprotective properties are mostly due to inhibiting the expression of adhesion molecules, thereby reducing the adherence of monocytes to endothelial cells. In addition, adiponectin reduces plaque formation and increases plaque stability and nitric oxide production [154].

The adiponectin signal is transmitted through the ADIPOR1 and ADIPOR2 receptors, followed by the docking of the adaptor protein APPL1 [155]. By activating the receptor, the signaling pathway leads to metabolic improvements, including a reduction in hepatic glycogenolysis [156]. This resulted in an increase in the oxidation of fatty acids in the liver and skeletal muscle. In addition, there was an increase in the uptake of glucose in both the skeletal muscle and WAT, and a reduction in inflammation in the WAT [157]. It is therefore clear that the skeletal muscle, liver, and adipose tissue are highly enriched with adiponectin receptors [158]. 

In addition, adiponectin receptors are found in the pancreas, where it inhibits fat storage and reduces pancreatic cancer development by neutralizing inflammatory ceramides and diacylglycerols [156]. The anti-inflammatory effects of adiponectin have been demonstrated to extend to other cell types, such as macrophages and fibrogenic cells [157,159,160]. This anti-inflammatory effect is due to suppressing the expression of IL-8 in tumor necrosis factor α (TNF-α)-stimulated human aortic endothelial cells [161]. It also suppresses the expression of TNFα and monocyte chemoattractant protein 1 in several cell types (including human circulating monocyte-derived macrophages, stromal vascular fraction cells from human subcutaneous fat pads, and murine alveolar macrophages) [153]. In contrast, however, in human monocyte-derived macrophages, adiponectin increases the expression of the anti-inflammatory cytokine IL-10 [162].

Recently, increasing adiponectin levels have been associated with a lower risk of developing diabetes across populations in a dose–response relationship; however, dysregulation of adiponectin has been implicated in obesity, metabolic syndrome, type 2 diabetes, hypertension, and cardiovascular disease [163]. Indeed, Liu et al. concluded that circulating adiponectin might serve as an available diagnostic biomarker to identify metabolic syndrome subjects, especially in high-risk populations with insulin resistance [164]. Finally, it appears that, similar to leptin, physical exercise exerts adiponectin-enhancing effects comparable to those of some anti-diabetic drugs [165]. In addition, Becic et al. reported that physical exercise in general, and aerobic exercise in particular, significantly increases adiponectin levels in prediabetic and diabetic adults [154].

### 3.3. Resistin

The RETN gene encodes a cysteine-rich polypeptide known as resistin [166]. An investigation of thiazolidinedione targets in the white adipose tissue of mice led to the discovery of this hormone, which plays a significant role in insulin resistance development [167].

As a result of the action of resistin, a hormone secreted by adipose tissue, glucose homeostasis is impaired in mice. It is this process that leads to the development of type 2 diabetes mellitus. Visceral obesity and diabetes are linked by resistin [167,168]. The circulatory system is abundant with resistin, a small secretory protein rich in cysteine and cysteine [169]. To meet the energy needs of the body, adipose tissue secretes FFA [170]. Additionally, it secretes several small polypeptides that are specific to adipose tissue, such as leptin, adiponectin, and resistin [170]. The expression of resistin in human adipose tissue is very low, but it is highly expressed in circulating mononuclear leukocytes and macrophages. These levels are reduced in obese and prediabetic individuals [170,171].

In a recent study, it was demonstrated that human resistin induces mitochondrial dysfunction through the abnormal fission of mitochondria. Considering these findings, the resistin–CAP1 complex may represent a potential therapeutic target for obesity-related metabolic diseases such as diabetes and cardiometabolic disease [172].

## 4. Exercise and Leptin Control in Children with Obesity

Overweight and obesity are consequences of excess caloric intake and/or low levels of energy expenditure [173]. However, the concomitant measurement of both total energy expenditure and resting energy expenditure in children with obesity concluded that reduced resting energy expenditure alone is not the primary cause of common obesity, but rather an alteration of the hypothalamic pathways that regulate energy expenditure [174]. At rest, energy expenditure is usually defined as that spent on metabolism to keep the organism alive [175]. All of these processes generate heat and are therefore thermogenic, with lectin being a key hormone [176].

According to the WHO, physical activity benefits the heart, body, and mind [177]. It is well known that physical activity contributes to the prevention and management of noncommunicable diseases such as cardiovascular disease, cancer, and diabetes [177]. Depression and anxiety can be reduced by physical activity. By engaging in physical activity, one improves one’s ability to think, learn, and judge [177]. In order for young people to grow and develop in a healthy manner, physical activity is essential. The benefits of physical activity extend to a better quality of life [177]. The global average for physical activity is not met by one in four adults [177]. An insufficient level of physical activity is associated with an increased risk of death by up to 30% when compared to an adequate level of activity [177]. Insufficient physical activity is a problem for more than 80% of the world’s adolescent population [177].

Based on the updated literature search, 21 systematic reviews were identified as relevant [178]. The evidence reviewed (i.e., systematic reviews published in the past and those that have been published recently) suggests that physical activity is associated with improved health outcomes (primarily intermediate outcomes) when it is increased in amount and intensity, and when it is undertaken in a variety of ways (e.g., aerobic exercises and muscle and bone strengthening exercises) [178]. In 2010, the WHO guidelines did not address the issue of limiting sedentary behaviors, which was supported by sufficient evidence. There is, however, a lack of evidence to fully describe the dose–response relationship between physical activity and health outcomes, or whether these associations are associated with specific types or domains of physical activity or sedentary behavior [178].

Recent studies show that compared to normal-weight children, children with obesity tend to have higher circulating leptin that decreases with decreasing BMI. This is due to insensitivity to circulating leptin and the disruption of leptin signaling in the hypothalamus [179,180]. Low levels of leptin, which can be linked to a low body weight, were associated with high physical activity in school children in rural Tanzania according to Ludwig et al. [181]. The child population is characterized by an active lifestyle, e.g., an active mode of traveling to school, playing outdoors during recess or after school, household and agricultural chores, and very low screen time. Indeed, the current study considered the daily number of steps (median of around 17,000 steps), with 57% of children reaching the limit of 16,500 steps per day [181]. Thus, these results indicate an endocrine pathway affecting physical activity levels. In addition, they have already shown signs of increased physical activity in children with higher leptin levels. Furthermore, García-Hermoso et al. believe that the longer the exercise time (≥60 min), the higher the energy expenditure (≥800 kcal) and the lower the leptin concentration, which indicates that the duration of exercise training and the time of exercise training are significantly negatively correlated with the leptin level [182]. Therefore, it could be suggested that evidence points to a role for the hormone leptin in modulating physical activity. 

Physical activity is already known to be the most effective non-pharmacological remedy to treat obesity or overweight because it increases energy expenditure, as well as helping to balance energy levels [183]. It has been found that Spanish children who participate in a total of seven hours of physical activity per week have significantly lower values of waist circumference, fat mass index, and homeostatic model assessment index [184]. The result of a novel study show that it is important to stimulate greater collaboration among family members and to increase exercise through extracurricular activities to promote the activity of children during the growth stage [185].

Indeed, in children, the regular practice of moderate to vigorous physical activity could improve endothelial function [186]. In their meta-analysis, Sirico et al. demonstrated that physical exercise alone, without concurrent dietary modification or other lifestyle changes, decreased plasma levels of leptin and IL-6, indicating a reduction in systemic inflammation associated with obesity [187]. Regarding exercise duration, the available studies reported mixed results, with insufficient data to draw conclusions about leptin’s response to different modes of exercise [187]. Although aerobic exercise is considered optimal for reducing body fat, resistance exercise leads to an increase in lean body mass [188]. Furthermore, Belcher et al. concluded that leptin values predicted a decline in physical activity during the start of puberty, independent of central adiposity. Indeed, it was found that girls with low leptin levels had higher levels of physical activity than girls with high leptin levels at the start of puberty [189]. Jimenez-Pavon et al. reported that vigorous physical activity and fitness moderate the levels of leptin concentrations, regardless of relevant confounders, including total body fat [190]. Hence, these results suggest that not only average physical activity but also high-intensity physical activity could influence leptin concentrations independently of physical fitness levels [190]. Likewise, Miyatake et al. described that the peak oxygen uptake in men and physical activity in women were closely associated with circulating leptin levels, even after adjusting for confounding factors [191]. Remmel et al. observed that serum leptin concentrations were higher in overweight individuals than in normal-weight individuals. In addition, we found that leptin was negatively correlated with total physical activity and positively with sedentary time [192].

Briefly, the most effective recommendation for the prevention of obesity is to maintain a healthy lifestyle through balanced nutrition and physical activity, but this may be impossible to adhere to once a person enters the circular reward–deficiency syndrome [193]. Therefore, future longitudinal studies are needed to clarify the role of physical activity in weight control in overweight children. Figure 1 summarizes the main findings of the present review.

## 5. Future Perspectives

Overweight and obesity in children and adolescents are among the most important health risks in the world [194]. One in five children and adolescents in the United States suffer from childhood obesity. All children are at a risk of gaining weight above what is considered healthy, but some groups of children are more affected than others [195]. A child’s obesity likelihood is twice as high (23.2%) if he or she comes from a low-income family as compared to a child from a high-income family (11.9%). The epidemic is more prevalent in Spanish schools located in districts with elevated child poverty rates [194,196].

In spite of the alarming statistics regarding childhood obesity, there is still time to improve the health of the subject. For this purpose, it is imperative to understand the global prevalence of child and adolescent obesity and recognize the behavioral factors that contribute to excessive weight gain in this age group. Nutrition and physical activity are critical factors to consider in the management of child and adolescent obesity. We can prevent childhood obesity by identifying a variety of strategies. During the diagnostic and treatment process, it is necessary to discuss challenges specific to the country and clinical scenarios that may be encountered. Therefore, physical exercise could be a critical strategy to control obesity and the progression of an inflammatory status in the pediatric population in relation to some adipocytokines. In fact, current studies on human models of obesity, diabetes, and atherosclerosis have reported the potential role of adiponectin and its receptors in these metabolic diseases. Since endogenous adiponectin production is impaired as an effect of obesity and related pathologies, a practical therapeutic approach of using pharmacological or dietary interventions to restore the ability of adipose tissue to secrete adiponectin could be an interesting strategy to use.

Based on the results presented, future research should follow three clear directions. The first is to elucidate the differential responses in aerobic capacity measures and hormonal responses based on the level of stimulation within a type of exercise. This will provide key information that might be able to provide insight into the possible response depending on the type of exercise proposed. Second, we must note that the various settings in which exercise and physical activity programs are provided are significant, from examining the role of the school system to be able to increase overall activity and the use of structured and self-chosen exercise programs, to incorporating these programs to motivate overweight children and adolescents. This will improve health and body composition, but also academic progress, based on the idea that increased physical activity leads to improved academic performance [197]. Finally, further examination of the effects of inactivity on physiological and hormonal homeostasis during youth is crucial, as 45% of children aged 6–11 years fail to achieve the recommended 60 min/day of moderate–vigorous physical activity, which is even lower in the adolescent population [198,199]. 

In this review, we explored some orchestrators involved in this health condition in pediatric populations and how they are interconnected. We could discover new key molecules, as well as detect significant factors, in the coming years if more randomized clinical trials with large sample sizes and long-term studies are conducted. By combining omics analyses and machine learning techniques, it is possible to improve treatment decisions for the pediatric population in order to improve their health.

## Figures and Tables

**Figure 1 ijms-23-15413-f001:**
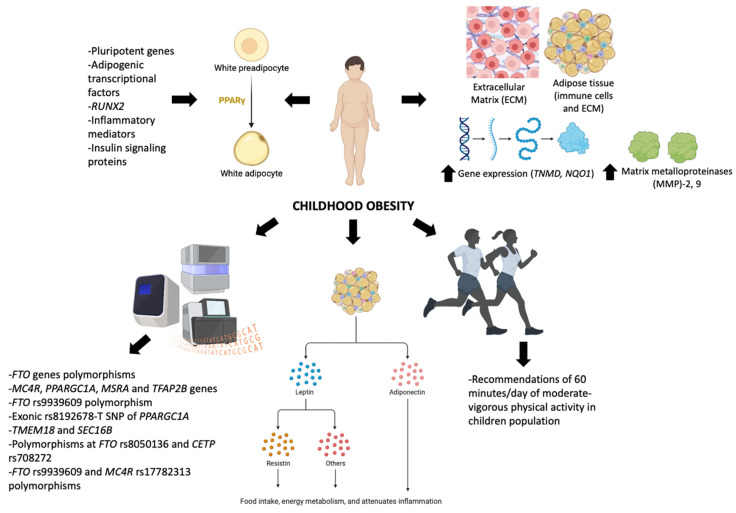
Children’s obesity and the role of molecular and hormonal factors.

## Data Availability

Not applicable.

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
