# Peer review of "The Role of Molecular and Hormonal Factors in Obesity and the Effects of Physical Activity in Children"

_ijms, 2022, doi:10.3390/ijms232315413_

Round 1

Reviewer 1 Report

The manuscript entitled “The role of molecular and hormonal factors in obesity and the effects of physical activity in children” could be interesting but the article merits attention.

Overall the description of molecular factors is a mere list rather than an intellectual digest and cohesive presentation of the quoted studies. This review summarizes and might stimulate research in the field. 

Literature used

The manuscript falls short regarding several quality criteria of reviews. The selection of original articles is somewhat arbitrary and often, important articles are not cited. Generally, the literature review strategy should be declared and where possible, reference to the initial original evidence and, eventually, to the following important contributions should be given.

Citing other reviews should be avoided unless they refer to aspects not reviewed in the present article.

The authors are invited to assume a more critical position on the literature cited and to discuss more extensively contradicting evidence.

In the abstract, the aim of the review is too general. Also, in section 5 authors declared “ In this review, we explored some orchestrators involved in this disease in pediatric

populations and how they are interconnected”, but the interconnection is not displayed.

Additionally, the authors emphasize “A larger number of randomized clinical trials with larger sample sizes and long-term studies could lead to the discovery of new key molecules as well as the detection of significant factors in the coming years”, but this was not accurately discussed in the manuscript.

Please discuss in the introduction the differences in criteria for childhood obesity and the controversy when BMI is used for obesity classification in children.

Separate section 2 into subtopics. It seems that it would be discussed according to the type of adipose tissue, but it is not clear the structure due to a lack of continuity.

Line 209 to 216, is very general. For example, high levels of pyruvate, lactate, and alanine are mentioned, but in which tissues? under what conditions? What are the high levels of these metabolites due to? Citing original research is required, not relying on reviews. Please review all cited references, because, for example, reference 77 was cited 3 times in the same paragraph.

The evidence of genes related to childhood obesity is very broad and differs between age ranges, please make a critical analysis. It is suggested to use a table.

Section 3

Also, divide this section into subtopics according to each hormone. Why only leptin and adiponectin were discussed?

Section 4. The figure is not representative of what has been reported, it is very general and the lower arrows indicate that physical activity, molecular control, and hormones lead to childhood obesity. It is suggested to add more figures or tables and improve this figure that does not give any specific information.

Section 5 needs to be improved with a critical analysis of the information reviewed.

Author Response

Dear Ms. Fia Cheng

Section Managing Editor,

Thank you for providing us with the opportunity to submit a revised version of our editorial entitled The role of molecular and hormonal factors in obesity and the effects of physical activity in childrento the International Journal of Molecular Sciences in the Special Issue “Frontiers in Obesity”.

We would like to thank the reviewers for their thoughtful comments and suggestions regarding our manuscript. All comments received from reviewers have been incorporated into the revised manuscript. Below is an itemized point-by-point response to the comments from the reviewers in response to the changes made to the original document (highlighted in yellow).

COMMENTS FROM REVIEWER 1

 Comment #1

The manuscript entitled “The role of molecular and hormonal factors in obesity and the effects of physical activity in children” could be interesting but the article merits attention. Overall, the description of molecular factors is a mere list rather than an intellectual digest and cohesive presentation of the quoted studies. This review summarizes and might stimulate research in the field. Literature used; the manuscript falls short regarding several quality criteria of reviews. The selection of original articles is somewhat arbitrary and often, important articles are not cited. Generally, the literature review strategy should be declared and where possible, reference to the initial original evidence and, eventually, to the following important contributions should be given.

Response: Thanks to the reviewer for his/her kind comment about our manuscript. Regarding the review articles, we have added original articles where ever possible. The literature review strategy was added in the manuscript and now states (page 3, lines 130-134), “A comprehensive literature search was conducted using electronic databases, including Medline (PubMed), EMBASE, and the Cochrane Library. The keywords we used were "obesity", "physical activity", "exercise", "hormones", "childhood obesity", "children" and "adipose tissue". For additional relevant literature, we searched the reference lists of the articles included in the review.”

Comment #2

Citing other reviews should be avoided unless they refer to aspects not reviewed in the present article. The authors are invited to assume a more critical position on the literature cited and to discuss more extensively contradicting evidence.

Response: Using the reviewer’s comment the use of reviews was limited where ever possible, and changes in the manuscript related to discuss more extensively contradicting evidence was added.

Comment #3

In the abstract, the aim of the review is too general. Also, in section 5 authors declared “In this review, we explored some orchestrators involved in this disease in pediatric populations and how they are interconnected”, but the interconnection is not displayed.

Response: Using the reviewer’s comment, the sentence was modified and now states (page 3, lines 135-138), “Therefore, the present review aims to investigate the molecular and hormonal mechanisms recently analyzed in obesity. In addition, the present review aims to investigate the involvement of leptin in physical exercise and its relevance in the control of obesity, especially in the children population.”

Comment #4

Additionally, the authors emphasize “A larger number of randomized clinical trials with larger sample sizes and long-term studies could lead to the discovery of new key molecules as well as the detection of significant factors in the coming years”, but this was not accurately discussed in the manuscript.

Response: Based on the reviewer's comment, more discussion has been added to complement the present review's conclusion.

Comment #5

Please discuss in the introduction the differences in criteria for childhood obesity and the controversy when BMI is used for obesity classification in children.

Response: Using the reviewer’s comment, we have added more information on this topic in the introduction and the manuscript now states (page 2, lines 45-64), “It is possible to calculate BMI in adults to determine obesity; however, it is more difficult in children to determine it accurately [5]. Overweight and childhood obesity are currently classified according to three classification systems. They include the criteria published in 2007 by the World Health Organization (WHO) [6] the data tables published by the Centers for Disease Control and Prevention (CDC) in 2000 [7] and the guidelines published by the International Obesity Task Force (IOTF or Cole-IOTF) [8]. According to the WHO, overweight is considered as body weight greater than one standard deviation above the median of the WHO reference growth standard. Generally, obesity is classified as weight over 2 standard deviations above the median of the WHO growth reference standard for height [6]. Overweight is defined by the CDC as BMI exceeding the 85th percentile but below the 95th percentile. Obesity is defined as BMI exceeding the 95th percentile, according to age and gender [7]. Using the cut-offs specified for adults as a basis, Cole et al developed BMI curves identifying overweight as (25-30 kg/m2) and obesity as (>30 kg/m2) by age group and gender [8]. Because of its simplicity and low cost, BMI has become an increasingly popular indicator for identifying overweight and obesity [9]. BMI does not distinguish between increased mass in the form of fat, lean tissue or bone, and consequently, it may lead to significant misclassification, especially in children and adolescents [10]. Consequently, other electrical bioimpedance measurements are required in order to provide a more complete estimate of additional body compartments [11,12].

Comment #6

Separate section 2 into subtopics. It seems that it would be discussed according to the type of adipose tissue, but it is not clear the structure due to a lack of continuity.

Response: Using the reviewer’s comment, we have divided the section 2 into three different subtopics, the first related to “Structure and components of ECM related to obesity”, the second for “Control of differentiation in WAT “and the last one related to “Genetics underlying childhood obesity”

Comment #7

Line 209 to 216, is very general. For example, high levels of pyruvate, lactate, and alanine are mentioned, but in which tissues? under what conditions? What are the high levels of these metabolites due to? Citing original research is required, not relying on reviews. Please review all cited references, because, for example, reference 77 was cited 3 times in the same paragraph.

Response: Using the reviewer’s comment, we have added more information on this topic and the manuscript now states (page 6, lines 278-286), “High levels of pyruvate, lactate, and alanine were detected in resting blood samples, indicating altered metabolism in the early years of a child's life [102-104]. Elevated pyruvate levels suggest a deficiency in the enzyme pyruvate dehydrogenase (complex), which is needed to make acetyl-CoA going forward [101]. Regarding lactate increases, it could show deregulations in central carbon metabolism and tend to direct metabolism towards fermentation conditions in children with obesity, which is called "aerobic glycolysis" or "Warburg effect" [105]. Therefore, from a physiological point of view, obesity corresponds hand in hand with adipocyte hypertrophy that is associated with local hypoxia that enhances lactate production [106].”

Comment #8

The evidence of genes related to childhood obesity is very broad and differs between age ranges, please make a critical analysis. It is suggested to use a table.

Response: Using the reviewer’s comment, we have added more information on this topic and the manuscript now states (pages 6-7, lines 291-340),” Studies of cross-sectional observational design strongly suggest that childhood obesity or childhood leanness is associated with certain family lines. The weight status of parents is strongly correlated with the weight status of their offspring up to the age of five [108]. In addition, it is associated with the risk of obesity in adulthood. There is an association between birth weight and the risk of obesity later in life, with low and high birth weight babies being at greatest risk [108].

Studies of twins, families, and adoptions have estimated the heritability of obesity to range between 40% and 70% [109-111]. In line with this fact, genetic approaches can be utilized in order to characterize the physiological and molecular mechanisms that are responsible for controlling weight [109-111]. However, the evidence of genes related to childhood obesity is very broad and sometimes differs between age ranges. For this reason, Table 1 summarizes the main variables of the included studies that dealt with genetics and the onset of childhood obesity.

The authors of a meta-analysis published in 2010 found that genetic factors play an influential role in the variation of BMI at all ages. A substantial effect was also seen in middle childhood by common environmental factors, but this effect disappeared in adolescence [112].

Methylation of the long non-coding RNA ANRIL (encoded at CDKN2A) was associated with adiposity in birth tissues of ethnically diverse neonates, peripheral blood of adolescents, and adipose tissue of adults. Perinatal methylation at gene function loci may serve as a robust indicator of later adiposity [113].

A total of ten polymorphisms were evaluated in 730 Portuguese children ages 6 to 12 in order to assess their vulnerability to obesity. Methionine sulfoxide reductase A (MSRA), transcription factor AP-2 beta (TFAP2B), melanocortin 4 receptor (MC4R), neurexin 3 (NRXN3), peroxisome proliferator-activated receptor gamma coactivator 1 alpha (PPARGC1A), transmembrane protein 18 (TMEM18), homolog of Sec16 (SEC16B), homeobox B5 (HOXB5), and olfactomedin 4 (OLFM4) were evaluated [114]. The results of this study suggest that polymorphisms of the MC4R, PPARGC1A, MSRA, and TFAP2B genes may be associated with obesity-related traits in a sample of Portuguese children [114].

Based on a meta-analysis that included 12 eligible studies with 5,000 cases and 9,853 controls, the FTO rs9939609 polymorphism was significantly associated with an increased risk of obesity [115].

According to the meta-analysis of ALSPAC and Raine samples, a novel sin-gle-nucleotide polymorphisms (SNP) was detected downstream of the FAM120AOS gene on chromosome 9 [116]. The association was triggered by differences in BMI at 8 years (T allele of rs944990 increased BMI with a modest association with change in BMI over time) [116]. A locus associated with childhood obesity (OLFM4) has reached genome-wide significance in relation to BMI at age 8 and/or changes over time [116].

In addition, there is a strong association between the exonic rs8192678-T SNP of PPARGC1A and a reduction in BMI z-score [117]. Another study has shown that SEC16B and TMEM18 were associated with 27% and 40% increased odds of obesity, respectively, in Hispanic/Latinos children (22-88% frequency) [118].

According to the IDEFICS/I.Family study, significant associations were found for 5 SNPs for the FTO and CETP genes [119].

Several studies have shown a significant association between FTO rs9930506 and MC4R rs17782313 polymorphisms and obesity in children [120]. These genetic variants were associated with childhood obesity in Caucasians and Asians, according to stratified analyses [120]. According to a recent systematic review, the polymorphisms rs9939609 FTO and rs17782313 MC4R could be associated with overweight and obesity in children and adolescents. This depends on the study population and ethnicity [121].”Also a Table (page 8) was added with the main information.

Comment #9

Section 3

Also, divide this section into subtopics according to each hormone. Why only leptin and adiponectin were discussed?

Response: Using the reviewer’s comment, we have divided section 3 according to hormones and added information about leptin and adiponectin and also resistin was added to the manuscript and now states (pages 11-12, lines 519-535), “3.3. Resistin

The RETN gene encodes a cysteine-rich polypeptide known as resistin [160]. An investigation of thiazolidinedione targets in white adipose tissue of mice led to the discovery of this hormone, which plays a significant role in insulin resistance development [161].

As a result of the action of resistin, a hormone secreted by adipose tissue, glucose homeostasis is impaired in mice. It is this process that leads to the development of type 2 diabetes mellitus. Visceral obesity and diabetes are linked by resistin [161,162]. The circulatory system is abundant with resistin, a small secretory protein rich in cysteine and cysteine [163]. To meet the energy needs of the body, adipose tissues secrete FFA [164]. Additionally, it secretes several small polypeptides that are specific to adipose tissue, such as leptin, adiponectin, and resistin [164]. The expression of resistin in human adipose tissue is very low, but it is highly expressed in circulating mononuclear leukocytes and macrophages. These levels are reduced in obese and prediabetic individuals [164,165].

In a recent study, it was demonstrated that human resistin induces mitochondrial dysfunction through abnormal fission of mitochondria. Considering these findings, the resistin-CAP1 complex may represent a potential therapeutic target for obesity-related metabolic diseases such as diabetes and cardiometabolic disease [166].

Comment #10

Section 4. The figure is not representative of what has been reported, it is very general and the lower arrows indicate that physical activity, molecular control, and hormones lead to childhood obesity. It is suggested to add more figures or tables and improve this figure that does not give any specific information.

Response: Using the reviewer’s comment, we have modified the figure with specific information.

Comment #11

Section 5 needs to be improved with a critical analysis of the information reviewed.

Response: Using the reviewer’s comment, the section 5 was modified and now states (pages 14-15, lines 638-679), “Overweight and obesity in children and adolescents are among the most important health risks in the world [186]. One in five children and adolescents in the United States suffer from childhood obesity. All children are at risk of gaining weight above what is considered healthy, but some groups of children are more affected than others [187]. A child's obesity rate is twice as high (23.2%) if he or she comes from a low-income family as compared to a child from a high-income family (11.9%). The epidemic is more prevalent in Spanish schools located in districts with elevated child poverty rates [186,188].

In spite of the alarming statistics regarding childhood obesity, there is still time to improve the health of the subject. For this purpose, it is imperative to understand the global prevalence of child and adolescent obesity. Recognize the behavioral factors that contribute to excessive weight gain in this age group. Nutrition and physical activity are critical factors to consider in the management of child and adolescent obesity. Prevent childhood obesity by identifying a variety of strategies. During the diagnostic and treatment process, discuss challenges specific to the country and clinical scenarios that may be encountered. Therefore, physical exercise could be a critical strategy to control obesity and the progression of inflammatory status in the pediatric population in relation to some adipocytokines. In fact, current studies in human models of obesity, diabetes and atherosclerosis have reported the potential role of adiponectin and its receptors in these metabolic diseases. Since endogenous adiponectin production is impaired as an effect of obesity and related pathologies, a practical therapeutic approach of using pharmacological or dietary interventions to restore the ability of adipose tissue to secrete adiponectin could be an interesting strategy to use.

Based on the results presented, future research should follow three clear directions. First, to elucidate the differential responses in aerobic capacity measures and hormonal responses based on the level of stimulation within a type of exercise. This will provide key information that might be able to give insight into the possible response depending on the type of exercise proposed. Second, we cannot forget that the various settings in which exercise and physical activity programs are provided are significant, from examining the role of the school system to be able to increase overall activity and the use of structured and self-chosen exercise programs to incorporating these programs to motivate overweight children and adolescents. This will improve health and body composition, but also academic progress, based on the idea that increased physical activity leads to improved academic performance [189]. And finally, further examination of the effects of inactivity on physiological and hormonal homeostasis during youth is crucial, as 45% of children aged 6-11 years fail to achieve the recommended 60 minutes/day of moderate-vigorous physical activity, which is even lower in the adolescent population [190,191].

In this review, we explored some orchestrators involved in this disease in pediatric populations and how they are interconnected. We could discover new key molecules as well as detect significant factors in the coming years if more randomized clinical trials with large sample sizes and long-term studies are conducted. By combining omics analyses and machine learning techniques, it is possible to improve treatment decisions for the pediatric population in order to improve their health.”

Reviewer 2 Report

This paper dont show original datas, is very similar as other papers before published.

It would be accept only is a invitated paper one

You can find other similar papers

1. PMID: 32370020 Cluster Analysis of Physical Activity Patterns, and Relationship with Sedentary Behavior and Healthy Lifestyles in Prepubertal Children: Genobox Cohort. Leis R, Jurado-Castro JM, Llorente-Cantarero FJ, Anguita-Ruiz A, Iris-Rupérez A, Bedoya-Carpente JJ, Vázquez-Cobela R, Aguilera CM, Bueno G, Gil-Campos M.Nutrients. 2020 May 1;12(5):1288. doi: 10.3390/nu12051288.PMID: 32370020 Free PMC article. Remove from clipboard  Cite   Share 2. PMID: 34205732 Associations of MC4RLEP, and LEPR Polymorphisms with Obesity-Related Parameters in Childhood and Adulthood. Raskiliene A, Smalinskiene A, Kriaucioniene V, Lesauskaite V, Petkeviciene J.Genes (Basel). 2021 Jun 21;12(6):949. doi: 10.3390/genes12060949.PMID: 34205732 Free PMC article. Remove from clipboard  Cite   Share 3. PMID: 25673413 Genetic studies of body mass index yield new insights for obesity biology. Locke AE, Kahali B, Berndt SI, Justice AE, Pers TH, Day FR, Powell C, Vedantam S, Buchkovich ML, Yang J, Croteau-Chonka DC, Esko T, Fall T, Ferreira T, Gustafsson S, Kutalik Z, Luan J, Mägi R, Randall JC, Winkler TW, Wood AR, Workalemahu T, Faul JD, Smith JA, Zhao JH, Zhao W, Chen J, Fehrmann R, Hedman ÅK, Karjalainen J, Schmidt EM, Absher D, Amin N, Anderson D, Beekman M, Bolton JL, Bragg-Gresham JL, Buyske S, Demirkan A, Deng G, Ehret GB, Feenstra B, Feitosa MF, Fischer K, Goel A, Gong J, Jackson AU, Kanoni S, Kleber ME, Kristiansson K, Lim U, Lotay V, Mangino M, Leach IM, Medina-Gomez C, Medland SE, Nalls MA, Palmer CD, Pasko D, Pechlivanis S, Peters MJ, Prokopenko I, Shungin D, Stančáková A, Strawbridge RJ, Sung YJ, Tanaka T, Teumer A, Trompet S, van der Laan SW, van Setten J, Van Vliet-Ostaptchouk JV, Wang Z, Yengo L, Zhang W, Isaacs A, Albrecht E, Ärnlöv J, Arscott GM, Attwood AP, Bandinelli S, Barrett A, Bas IN, Bellis C, Bennett AJ, Berne C, Blagieva R, Blüher M, Böhringer S, Bonnycastle LL, Böttcher Y, Boyd HA, Bruinenberg M, Caspersen IH, Chen YI, Clarke R, Daw EW, de Craen AJM, Delgado G, Dimitriou M, Doney ASF, Eklund N, Estrada K, Eury E, Folkersen L, Fraser RM, Garcia ME, Geller F, Giedraitis V, Gigante B, Go AS, Golay A, Goodall AH, Gordon SD, Gorski M, Grabe HJ, Grallert H, Grammer TB, Gräßler J, Grönberg H, Groves CJ, Gusto G, Haessler J, Hall P, Haller T, Hallmans G, Hartman CA, Hassinen M, Hayward C, Heard-Costa NL, Helmer Q, Hengstenberg C, Holmen O, Hottenga JJ, James AL, Jeff JM, Johansson Å, Jolley J, Juliusdottir T, Kinnunen L, Koenig W, Koskenvuo M, Kratzer W, Laitinen J, Lamina C, Leander K, Lee NR, Lichtner P, Lind L, Lindström J, Lo KS, Lobbens S, Lorbeer R, Lu Y, Mach F, Magnusson PKE, Mahajan A, McArdle WL, McLachlan S, Menni C, Merger S, Mihailov E, Milani L, Moayyeri A, Monda KL, Morken MA, Mulas A, Müller G, Müller-Nurasyid M, Musk AW, Nagaraja R, Nöthen MM, Nolte IM, Pilz S, Rayner NW, Renstrom F, Rettig R, Ried JS, Ripke S, Robertson NR, Rose LM, Sanna S, Scharnagl H, Scholtens S, Schumacher FR, Scott WR, Seufferlein T, Shi J, Smith AV, Smolonska J, Stanton AV, Steinthorsdottir V, Stirrups K, Stringham HM, Sundström J, Swertz MA, Swift AJ, Syvänen AC, Tan ST, Tayo BO, Thorand B, Thorleifsson G, Tyrer JP, Uh HW, Vandenput L, Verhulst FC, Vermeulen SH, Verweij N, Vonk JM, Waite LL, Warren HR, Waterworth D, Weedon MN, Wilkens LR, Willenborg C, Wilsgaard T, Wojczynski MK, Wong A, Wright AF, Zhang Q; LifeLines Cohort Study, Brennan EP, Choi M, Dastani Z, Drong AW, Eriksson P, Franco-Cereceda A, Gådin JR, Gharavi AG, Goddard ME, Handsaker RE, Huang J, Karpe F, Kathiresan S, Keildson S, Kiryluk K, Kubo M, Lee JY, Liang L, Lifton RP, Ma B, McCarroll SA, McKnight AJ, Min JL, Moffatt MF, Montgomery GW, Murabito JM, Nicholson G, Nyholt DR, Okada Y, Perry JRB, Dorajoo R, Reinmaa E, Salem RM, Sandholm N, Scott RA, Stolk L, Takahashi A, Tanaka T, van 't Hooft FM, Vinkhuyzen AAE, Westra HJ, Zheng W, Zondervan KT; ADIPOGen Consortium; AGEN-BMI Working Group; CARDIOGRAMplusC4D Consortium; CKDGen Consortium; GLGC; ICBP; MAGIC Investigators; MuTHER Consortium; MIGen Consortium; PAGE Consortium; ReproGen Consortium; GENIE Consortium; International Endogene Consortium, Heath AC, Arveiler D, Bakker SJL, Beilby J, Bergman RN, Blangero J, Bovet P, Campbell H, Caulfield MJ, Cesana G, Chakravarti A, Chasman DI, Chines PS, Collins FS, Crawford DC, Cupples LA, Cusi D, Danesh J, de Faire U, den Ruijter HM, Dominiczak AF, Erbel R, Erdmann J, Eriksson JG, Farrall M, Felix SB, Ferrannini E, Ferrières J, Ford I, Forouhi NG, Forrester T, Franco OH, Gansevoort RT, Gejman PV, Gieger C, Gottesman O, Gudnason V, Gyllensten U, Hall AS, Harris TB, Hattersley AT, Hicks AA, Hindorff LA, Hingorani AD, Hofman A, Homuth G, Hovingh GK, Humphries SE, Hunt SC, Hyppönen E, Illig T, Jacobs KB, Jarvelin MR, Jöckel KH, Johansen B, Jousilahti P, Jukema JW, Jula AM, Kaprio J, Kastelein JJP, Keinanen-Kiukaanniemi SM, Kiemeney LA, Knekt P, Kooner JS, Kooperberg C, Kovacs P, Kraja AT, Kumari M, Kuusisto J, Lakka TA, Langenberg C, Marchand LL, Lehtimäki T, Lyssenko V, Männistö S, Marette A, Matise TC, McKenzie CA, McKnight B, Moll FL, Morris AD, Morris AP, Murray JC, Nelis M, Ohlsson C, Oldehinkel AJ, Ong KK, Madden PAF, Pasterkamp G, Peden JF, Peters A, Postma DS, Pramstaller PP, Price JF, Qi L, Raitakari OT, Rankinen T, Rao DC, Rice TK, Ridker PM, Rioux JD, Ritchie MD, Rudan I, Salomaa V, Samani NJ, Saramies J, Sarzynski MA, Schunkert H, Schwarz PEH, Sever P, Shuldiner AR, Sinisalo J, Stolk RP, Strauch K, Tönjes A, Trégouët DA, Tremblay A, Tremoli E, Virtamo J, Vohl MC, Völker U, Waeber G, Willemsen G, Witteman JC, Zillikens MC, Adair LS, Amouyel P, Asselbergs FW, Assimes TL, Bochud M, Boehm BO, Boerwinkle E, Bornstein SR, Bottinger EP, Bouchard C, Cauchi S, Chambers JC, Chanock SJ, Cooper RS, de Bakker PIW, Dedoussis G, Ferrucci L, Franks PW, Froguel P, Groop LC, Haiman CA, Hamsten A, Hui J, Hunter DJ, Hveem K, Kaplan RC, Kivimaki M, Kuh D, Laakso M, Liu Y, Martin NG, März W, Melbye M, Metspalu A, Moebus S, Munroe PB, Njølstad I, Oostra BA, Palmer CNA, Pedersen NL, Perola M, Pérusse L, Peters U, Power C, Quertermous T, Rauramaa R, Rivadeneira F, Saaristo TE, Saleheen D, Sattar N, Schadt EE, Schlessinger D, Slagboom PE, Snieder H, Spector TD, Thorsteinsdottir U, Stumvoll M, Tuomilehto J, Uitterlinden AG, Uusitupa M, van der Harst P, Walker M, Wallaschofski H, Wareham NJ, Watkins H, Weir DR, Wichmann HE, Wilson JF, Zanen P, Borecki IB, Deloukas P, Fox CS, Heid IM, O'Connell JR, Strachan DP, Stefansson K, van Duijn CM, Abecasis GR, Franke L, Frayling TM, McCarthy MI, Visscher PM, Scherag A, Willer CJ, Boehnke M, Mohlke KL, Lindgren CM, Beckmann JS, Barroso I, North KE, Ingelsson E, Hirschhorn JN, Loos RJF, Speliotes EK.Nature. 2015 Feb 12;518(7538):197-206. doi: 10.1038/nature14177.PMID: 25673413 Free PMC article. Remove from clipboard  Cite   Share 4. PMID: 33903722 GHS-R suppression in adipose tissues protects against obesity and insulin resistance by regulating adipose angiogenesis and fibrosis. Lee JH, Fang C, Li X, Wu CS, Noh JY, Ye X, Chapkin RS, Sun K, Sun Y.Int J Obes (Lond). 2021 Jul;45(7):1565-1575. doi: 10.1038/s41366-021-00820-7. Epub 2021 Apr 26.PMID: 33903722 Free PMC article. Remove from clipboard  Cite   Share 5. PMID: 28511904 Analysis of the association of leptin and adiponectin concentrations with metabolic syndrome in children: Results from the IDEFICS study. Nappo A, González-Gil EM, Ahrens W, Bammann K, Michels N, Moreno LA, Kourides Y, Iacoviello L, Mårild S, Fraterman A, Molnàr D, Veidebaum T, Siani A, Russo P.Nutr Metab Cardiovasc Dis. 2017 Jun;27(6):543-551. doi: 10.1016/j.numecd.2017.04.003. Epub 2017 Apr 19.PMID: 28511904 Remove from clipboard  Cite   Share 6. PMID: 28794174 Maternal Western diet age-specifically alters female offspring voluntary physical activity and dopamine- and leptin-related gene expression. Ruegsegger GN, Grigsby KB, Kelty TJ, Zidon TM, Childs TE, Vieira-Potter VJ, Klinkebiel DL, Matheny M, Scarpace PJ, Booth FW.FASEB J. 2017 Dec;31(12):5371-5383. doi: 10.1096/fj.201700389R. Epub 2017 Aug 9.PMID: 28794174 Remove from clipboard  Cite   Share 7. PMID: 28768569 Tissue cell stress response to obesity and its interaction with late gestation diet. Saroha V, Dellschaft NS, Keisler DH, Gardner DS, Budge H, Sebert SP, Symonds ME.Reprod Fertil Dev. 2018 Mar;30(3):430-441. doi: 10.1071/RD16494.PMID: 28768569 Remove from clipboard  Cite   Share 8. PMID: 28476142 Dipeptidyl peptidase-4 (DPP-4) inhibition with linagliptin reduces western diet-induced myocardial TRAF3IP2 expression, inflammation and fibrosis in female mice. Aroor AR, Habibi J, Kandikattu HK, Garro-Kacher M, Barron B, Chen D, Hayden MR, Whaley-Connell A, Bender SB, Klein T, Padilla J, Sowers JR, Chandrasekar B, DeMarco VG.Cardiovasc Diabetol. 2017 May 5;16(1):61. doi: 10.1186/s12933-017-0544-4.PMID: 28476142 Free PMC article.

Author Response

Dear Ms. Fia Cheng

Section Managing Editor,

Thank you for providing us with the opportunity to submit a revised version of our editorial entitled The role of molecular and hormonal factors in obesity and the effects of physical activity in childrento the International Journal of Molecular Sciences in the Special Issue “Frontiers in Obesity”.

We would like to thank the reviewers for their thoughtful comments and suggestions regarding our manuscript. All comments received from reviewers have been incorporated into the revised manuscript. Below is an itemized point-by-point response to the comments from the reviewers in response to the changes made to the original document (highlighted in yellow).

COMMENTS FROM REVIEWER 2

Comment #1

This paper dont show original datas, is very similar as other papers before published. It would be accept only is a invitated paper one. You can find other similar papers:

-PMID: 32370020 Cluster Analysis of Physical Activity Patterns, and Relationship with Sedentary Behavior and Healthy Lifestyles in Prepubertal Children: Genobox Cohort. Leis R, Jurado-Castro JM, Llorente-Cantarero FJ, Anguita-Ruiz A, Iris-Rupérez A, Bedoya-Carpente JJ, Vázquez-Cobela R, Aguilera CM, Bueno G, Gil-Campos M. Nutrients. 2020 May 1;12(5):1288. doi: 10.3390/nu12051288.

-PMID: 34205732 Associations of MC4R, LEP, and LEPR Polymorphisms with Obesity-Related Parameters in Childhood and Adulthood. Raskiliene A, Smalinskiene A, Kriaucioniene V, Lesauskaite V, Petkeviciene J. Genes (Basel). 2021 Jun 21;12(6):949. doi: 10.3390/genes12060949.

-PMID: 25673413 Genetic studies of body mass index yield new insights for obesity biology. Locke AE, Kahali B, Berndt SI, Justice AE, Pers TH, Day FR, Powell C, Vedantam S, Buchkovich ML, Yang J, Croteau-Chonka DC, Esko T, Fall T, Ferreira T, Gustafsson S, Kutalik Z, Luan J, Mägi R, et al. Nature. 2015 Feb 12;518(7538):197-206. doi: 10.1038/nature14177.

-PMID: 33903722 GHS-R suppression in adipose tissues protects against obesity and insulin resistance by regulating adipose angiogenesis and fibrosis. Lee JH, Fang C, Li X, Wu CS, Noh JY, Ye X, Chapkin RS, Sun K, Sun Y. Int J Obes (Lond). 2021 Jul;45(7):1565-1575. doi: 10.1038/s41366-021-00820-7. Epub 2021 Apr 26.

-PMID: 28511904 Analysis of the association of leptin and adiponectin concentrations with metabolic syndrome in children: Results from the IDEFICS study. Nappo A, González-Gil EM, Ahrens W, Bammann K, Michels N, Moreno LA, Kourides Y, Iacoviello L, Mårild S, Fraterman A, Molnàr D, Veidebaum T, Siani A, Russo P. Nutr Metab Cardiovasc Dis. 2017 Jun;27(6):543-551. doi: 10.1016/j.numecd.2017.04.003. Epub 2017 Apr 19.

-PMID: 28794174 Maternal Western diet age-specifically alters female offspring voluntary physical activity and dopamine- and leptin-related gene expression. Ruegsegger GN, Grigsby KB, Kelty TJ, Zidon TM, Childs TE, Vieira-Potter VJ, Klinkebiel DL, Matheny M, Scarpace PJ, Booth FW. FASEB J. 2017 Dec;31(12):5371-5383. doi: 10.1096/fj.201700389R. Epub 2017 Aug 9.

-PMID: 28768569 Tissue cell stress response to obesity and its interaction with late gestation diet. Saroha V, Dellschaft NS, Keisler DH, Gardner DS, Budge H, Sebert SP, Symonds ME. Reprod Fertil Dev. 2018 Mar;30(3):430-441. doi: 10.1071/RD16494.

-PMID: 28476142 Dipeptidyl peptidase-4 (DPP-4) inhibition with linagliptin reduces western diet-induced myocardial TRAF3IP2 expression, inflammation and fibrosis in female mice. Aroor AR, Habibi J, Kandikattu HK, Garro-Kacher M, Barron B, Chen D, Hayden MR, Whaley-Connell A, Bender SB, Klein T, Padilla J, Sowers JR, Chandrasekar B, DeMarco VG. Cardiovasc Diabetol. 2017 May 5;16(1):61. doi: 10.1186/s12933-017-0544-4. PMID: 28476142 Free PMC article.

Response: Please accept our sincere apologies for the opinion expressed by the reviewer regarding the sameness and lack of novelty of our review. The purpose of this review is to examine the molecular and hormonal mechanisms recently studied in the context of obesity. The present review also examines the role of leptin in physical exercise and its relevance in the control of obesity, especially in children. Accordingly, the review was divided into i) molecular control of obesity (Structure and components of ECM related to obesity; Control of differentiation in WAT and Genetics underlying childhood obesity), ii) Hormonal control in obesity (Leptin, Adiponectin, and Resistin), iii) Exercise and leptin control in children with obesity and iv) Future perspectives. All the topics were focused on the children population. The manuscript contains some references that are listed for the reviewer in order to increase the information provided to him/her.

Round 2

Reviewer 1 Report

The reviewed version of the manuscript was successfully modified. I only have some suggestions for Table 1.

Table 1 needs to be improved. Please, specify for each row the type of polymorphisms (SNP, deletion, insertion, tandem repeat). 

Please review the last 3 rows: For example: "Polymorphisms at FTO rs8050136 and CETP rs708272 have been identified as significant" Is there an association with childhood obesity or BMI?? 

In the last row: "polymorphisms could be associated"... Is there an association of the SNPs or not?

Author Response

Dear Ms. Fia Cheng

Section Managing Editor,

Thank you for providing us with the opportunity to submit a revised version of our editorial entitled The role of molecular and hormonal factors in obesity and the effects of physical activity in childrento the International Journal of Molecular Sciences in the Special Issue “Frontiers in Obesity”.

We would like to thank the reviewers for their thoughtful comments and suggestions regarding our manuscript. All comments received from reviewers have been incorporated into the revised manuscript. Below is an itemized point-by-point response to the comments from the reviewers in response to the changes made to the original document (highlighted in yellow).

COMMENTS FROM REVIEWER 1

Comment #1

The reviewed version of the manuscript was successfully modified. I only have some suggestions for Table 1. Table 1 needs to be improved. Please, specify for each row the type of polymorphisms (SNP, deletion, insertion, tandem repeat).

Response: Thanks to the reviewer for his/her kind comment about our manuscript and revision. Using the reviewer’s comment, more information was added in Table 1 (page 8, lines 365-367).

Comment #2

Please review the last 3 rows: For example: "Polymorphisms at FTO rs8050136 and CETP rs708272 have been identified as significant" Is there an association with childhood obesity or BMI??

Response: Using the reviewer’s comment the information of Table 1 was revised and now states in Table 1 (page 8, lines 365-367), “Polymorphisms at FTO rs8050136 and CETP rs708272 have been identified as significant with childhood metabolic syndrome”

Comment #3

In the last row: "polymorphisms could be associated"... Is there an association of the SNPs or not?

Response: Using the reviewer’s comment the information of Table 1 was revised and now states in Table 1 (page 8, lines 365-367), “FTO rs9939609 and MC4R rs17782313 polymorphisms have been associated with overweight and obesity in children

Reviewer 2 Report

This is a good  paper

Only add these references more and try to xpalin them at discussion topics

Genetic studies of body mass index yield new insights for obesity biology. Locke AE, Kahali B, Berndt SI, Justice AE, Pers TH, Day FR, Powell C, Vedantam S, Buchkovich ML, Yang J, Croteau-Chonka DC, Esko T, Fall T, Ferreira T, Gustafsson S, Kutalik Z, Luan J, Mägi R, Randall JC, Winkler TW, Wood AR, Workalemahu T, Faul JD, Smith JA, Zhao JH, Zhao W, Chen J, Fehrmann R, Hedman ÅK, Karjalainen J, Schmidt EM, Absher D, Amin N, Anderson D, Beekman M, Bolton JL, Bragg-Gresham JL, Buyske S, Demirkan A, Deng G, Ehret GB, Feenstra B, Feitosa MF, Fischer K, Goel A, Gong J, Jackson AU, Kanoni S, Kleber ME, Kristiansson K, Lim U, Lotay V, Mangino M, Leach IM, Medina-Gomez C, Medland SE, Nalls MA, Palmer CD, Pasko D, Pechlivanis S, Peters MJ, Prokopenko I, Shungin D, Stančáková A, Strawbridge RJ, Sung YJ, Tanaka T, Teumer A, Trompet S, van der Laan SW, van Setten J, Van Vliet-Ostaptchouk JV, Wang Z, Yengo L, Zhang W, Isaacs A, Albrecht E, Ärnlöv J, Arscott GM, Attwood AP, Bandinelli S, Barrett A, Bas IN, Bellis C, Bennett AJ, Berne C, Blagieva R, Blüher M, Böhringer S, Bonnycastle LL, Böttcher Y, Boyd HA, Bruinenberg M, Caspersen IH, Chen YI, Clarke R, Daw EW, de Craen AJM, Delgado G, Dimitriou M, Doney ASF, Eklund N, Estrada K, Eury E, Folkersen L, Fraser RM, Garcia ME, Geller F, Giedraitis V, Gigante B, Go AS, Golay A, Goodall AH, Gordon SD, Gorski M, Grabe HJ, Grallert H, Grammer TB, Gräßler J, Grönberg H, Groves CJ, Gusto G, Haessler J, Hall P, Haller T, Hallmans G, Hartman CA, Hassinen M, Hayward C, Heard-Costa NL, Helmer Q, Hengstenberg C, Holmen O, Hottenga JJ, James AL, Jeff JM, Johansson Å, Jolley J, Juliusdottir T, Kinnunen L, Koenig W, Koskenvuo M, Kratzer W, Laitinen J, Lamina C, Leander K, Lee NR, Lichtner P, Lind L, Lindström J, Lo KS, Lobbens S, Lorbeer R, Lu Y, Mach F, Magnusson PKE, Mahajan A, McArdle WL, McLachlan S, Menni C, Merger S, Mihailov E, Milani L, Moayyeri A, Monda KL, Morken MA, Mulas A, Müller G, Müller-Nurasyid M, Musk AW, Nagaraja R, Nöthen MM, Nolte IM, Pilz S, Rayner NW, Renstrom F, Rettig R, Ried JS, Ripke S, Robertson NR, Rose LM, Sanna S, Scharnagl H, Scholtens S, Schumacher FR, Scott WR, Seufferlein T, Shi J, Smith AV, Smolonska J, Stanton AV, Steinthorsdottir V, Stirrups K, Stringham HM, Sundström J, Swertz MA, Swift AJ, Syvänen AC, Tan ST, Tayo BO, Thorand B, Thorleifsson G, Tyrer JP, Uh HW, Vandenput L, Verhulst FC, Vermeulen SH, Verweij N, Vonk JM, Waite LL, Warren HR, Waterworth D, Weedon MN, Wilkens LR, Willenborg C, Wilsgaard T, Wojczynski MK, Wong A, Wright AF, Zhang Q; LifeLines Cohort Study, Brennan EP, Choi M, Dastani Z, Drong AW, Eriksson P, Franco-Cereceda A, Gådin JR, Gharavi AG, Goddard ME, Handsaker RE, Huang J, Karpe F, Kathiresan S, Keildson S, Kiryluk K, Kubo M, Lee JY, Liang L, Lifton RP, Ma B, McCarroll SA, McKnight AJ, Min JL, Moffatt MF, Montgomery GW, Murabito JM, Nicholson G, Nyholt DR, Okada Y, Perry JRB, Dorajoo R, Reinmaa E, Salem RM, Sandholm N, Scott RA, Stolk L, Takahashi A, Tanaka T, van 't Hooft FM, Vinkhuyzen AAE, Westra HJ, Zheng W, Zondervan KT; ADIPOGen Consortium; AGEN-BMI Working Group; CARDIOGRAMplusC4D Consortium; CKDGen Consortium; GLGC; ICBP; MAGIC Investigators; MuTHER Consortium; MIGen Consortium; PAGE Consortium; ReproGen Consortium; GENIE Consortium; International Endogene Consortium, Heath AC, Arveiler D, Bakker SJL, Beilby J, Bergman RN, Blangero J, Bovet P, Campbell H, Caulfield MJ, Cesana G, Chakravarti A, Chasman DI, Chines PS, Collins FS, Crawford DC, Cupples LA, Cusi D, Danesh J, de Faire U, den Ruijter HM, Dominiczak AF, Erbel R, Erdmann J, Eriksson JG, Farrall M, Felix SB, Ferrannini E, Ferrières J, Ford I, Forouhi NG, Forrester T, Franco OH, Gansevoort RT, Gejman PV, Gieger C, Gottesman O, Gudnason V, Gyllensten U, Hall AS, Harris TB, Hattersley AT, Hicks AA, Hindorff LA, Hingorani AD, Hofman A, Homuth G, Hovingh GK, Humphries SE, Hunt SC, Hyppönen E, Illig T, Jacobs KB, Jarvelin MR, Jöckel KH, Johansen B, Jousilahti P, Jukema JW, Jula AM, Kaprio J, Kastelein JJP, Keinanen-Kiukaanniemi SM, Kiemeney LA, Knekt P, Kooner JS, Kooperberg C, Kovacs P, Kraja AT, Kumari M, Kuusisto J, Lakka TA, Langenberg C, Marchand LL, Lehtimäki T, Lyssenko V, Männistö S, Marette A, Matise TC, McKenzie CA, McKnight B, Moll FL, Morris AD, Morris AP, Murray JC, Nelis M, Ohlsson C, Oldehinkel AJ, Ong KK, Madden PAF, Pasterkamp G, Peden JF, Peters A, Postma DS, Pramstaller PP, Price JF, Qi L, Raitakari OT, Rankinen T, Rao DC, Rice TK, Ridker PM, Rioux JD, Ritchie MD, Rudan I, Salomaa V, Samani NJ, Saramies J, Sarzynski MA, Schunkert H, Schwarz PEH, Sever P, Shuldiner AR, Sinisalo J, Stolk RP, Strauch K, Tönjes A, Trégouët DA, Tremblay A, Tremoli E, Virtamo J, Vohl MC, Völker U, Waeber G, Willemsen G, Witteman JC, Zillikens MC, Adair LS, Amouyel P, Asselbergs FW, Assimes TL, Bochud M, Boehm BO, Boerwinkle E, Bornstein SR, Bottinger EP, Bouchard C, Cauchi S, Chambers JC, Chanock SJ, Cooper RS, de Bakker PIW, Dedoussis G, Ferrucci L, Franks PW, Froguel P, Groop LC, Haiman CA, Hamsten A, Hui J, Hunter DJ, Hveem K, Kaplan RC, Kivimaki M, Kuh D, Laakso M, Liu Y, Martin NG, März W, Melbye M, Metspalu A, Moebus S, Munroe PB, Njølstad I, Oostra BA, Palmer CNA, Pedersen NL, Perola M, Pérusse L, Peters U, Power C, Quertermous T, Rauramaa R, Rivadeneira F, Saaristo TE, Saleheen D, Sattar N, Schadt EE, Schlessinger D, Slagboom PE, Snieder H, Spector TD, Thorsteinsdottir U, Stumvoll M, Tuomilehto J, Uitterlinden AG, Uusitupa M, van der Harst P, Walker M, Wallaschofski H, Wareham NJ, Watkins H, Weir DR, Wichmann HE, Wilson JF, Zanen P, Borecki IB, Deloukas P, Fox CS, Heid IM, O'Connell JR, Strachan DP, Stefansson K, van Duijn CM, Abecasis GR, Franke L, Frayling TM, McCarthy MI, Visscher PM, Scherag A, Willer CJ, Boehnke M, Mohlke KL, Lindgren CM, Beckmann JS, Barroso I, North KE, Ingelsson E, Hirschhorn JN, Loos RJF, Speliotes EK.Nature. 2015 Feb 12;518(7538):197-206. doi: 10.1038/nature14177.PMID: 25673413 Free PMC article. Obesity is heritable and predisposes to many diseases. To understand the genetic basis of obesity better, here we conduct a genome-wide association study and Metabochip meta-analysis of body mass index (BMI), a measure commonly used to define obesity and asse … Item in Clipboard 2 Cite   Share   FTO Obesity Variant Circuitry and Adipocyte Browning in Humans. Claussnitzer M, Dankel SN, Kim KH, Quon G, Meuleman W, Haugen C, Glunk V, Sousa IS, Beaudry JL, Puviindran V, Abdennur NA, Liu J, Svensson PA, Hsu YH, Drucker DJ, Mellgren G, Hui CC, Hauner H, Kellis M.N Engl J Med. 2015 Sep 3;373(10):895-907. doi: 10.1056/NEJMoa1502214. Epub 2015 Aug 19.PMID: 26287746 Free PMC article. METHODS: We examined epigenomic data, allelic activity, motif conservation, regulator expression, and gene coexpression patterns, with the aim of dissecting the regulatory circuitry and mechanistic basis of the association between the FTO region and obesity. ...This …   3 Cite   Share   New genetic loci link adipose and insulin biology to body fat distribution. Shungin D, Winkler TW, Croteau-Chonka DC, Ferreira T, Locke AE, Mägi R, Strawbridge RJ, Pers TH, Fischer K, Justice AE, Workalemahu T, Wu JMW, Buchkovich ML, Heard-Costa NL, Roman TS, Drong AW, Song C, Gustafsson S, Day FR, Esko T, Fall T, Kutalik Z, Luan J, Randall JC, Scherag A, Vedantam S, Wood AR, Chen J, Fehrmann R, Karjalainen J, Kahali B, Liu CT, Schmidt EM, Absher D, Amin N, Anderson D, Beekman M, Bragg-Gresham JL, Buyske S, Demirkan A, Ehret GB, Feitosa MF, Goel A, Jackson AU, Johnson T, Kleber ME, Kristiansson K, Mangino M, Leach IM, Medina-Gomez C, Palmer CD, Pasko D, Pechlivanis S, Peters MJ, Prokopenko I, Stančáková A, Sung YJ, Tanaka T, Teumer A, Van Vliet-Ostaptchouk JV, Yengo L, Zhang W, Albrecht E, Ärnlöv J, Arscott GM, Bandinelli S, Barrett A, Bellis C, Bennett AJ, Berne C, Blüher M, Böhringer S, Bonnet F, Böttcher Y, Bruinenberg M, Carba DB, Caspersen IH, Clarke R, Daw EW, Deelen J, Deelman E, Delgado G, Doney AS, Eklund N, Erdos MR, Estrada K, Eury E, Friedrich N, Garcia ME, Giedraitis V, Gigante B, Go AS, Golay A, Grallert H, Grammer TB, Gräßler J, Grewal J, Groves CJ, Haller T, Hallmans G, Hartman CA, Hassinen M, Hayward C, Heikkilä K, Herzig KH, Helmer Q, Hillege HL, Holmen O, Hunt SC, Isaacs A, Ittermann T, James AL, Johansson I, Juliusdottir T, Kalafati IP, Kinnunen L, Koenig W, Kooner IK, Kratzer W, Lamina C, Leander K, Lee NR, Lichtner P, Lind L, Lindström J, Lobbens S, Lorentzon M, Mach F, Magnusson PK, Mahajan A, McArdle WL, Menni C, Merger S, Mihailov E, Milani L, Mills R, Moayyeri A, Monda KL, Mooijaart SP, Mühleisen TW, Mulas A, Müller G, Müller-Nurasyid M, Nagaraja R, Nalls MA, Narisu N, Glorioso N, Nolte IM, Olden M, Rayner NW, Renstrom F, Ried JS, Robertson NR, Rose LM, Sanna S, Scharnagl H, Scholtens S, Sennblad B, Seufferlein T, Sitlani CM, Smith AV, Stirrups K, Stringham HM, Sundström J, Swertz MA, Swift AJ, Syvänen AC, Tayo BO, Thorand B, Thorleifsson G, Tomaschitz A, Troffa C, van Oort FV, Verweij N, Vonk JM, Waite LL, Wennauer R, Wilsgaard T, Wojczynski MK, Wong A, Zhang Q, Zhao JH, Brennan EP, Choi M, Eriksson P, Folkersen L, Franco-Cereceda A, Gharavi AG, Hedman ÅK, Hivert MF, Huang J, Kanoni S, Karpe F, Keildson S, Kiryluk K, Liang L, Lifton RP, Ma B, McKnight AJ, McPherson R, Metspalu A, Min JL, Moffatt MF, Montgomery GW, Murabito JM, Nicholson G, Nyholt DR, Olsson C, Perry JR, Reinmaa E, Salem RM, Sandholm N, Schadt EE, Scott RA, Stolk L, Vallejo EE, Westra HJ, Zondervan KT; ADIPOGen Consortium; CARDIOGRAMplusC4D Consortium; CKDGen Consortium; GEFOS Consortium; GENIE Consortium; GLGC; ICBP; International Endogene Consortium; LifeLines Cohort Study; MAGIC Investigators; MuTHER Consortium; PAGE Consortium; ReproGen Consortium, Amouyel P, Arveiler D, Bakker SJ, Beilby J, Bergman RN, Blangero J, Brown MJ, Burnier M, Campbell H, Chakravarti A, Chines PS, Claudi-Boehm S, Collins FS, Crawford DC, Danesh J, de Faire U, de Geus EJ, Dörr M, Erbel R, Eriksson JG, Farrall M, Ferrannini E, Ferrières J, Forouhi NG, Forrester T, Franco OH, Gansevoort RT, Gieger C, Gudnason V, Haiman CA, Harris TB, Hattersley AT, Heliövaara M, Hicks AA, Hingorani AD, Hoffmann W, Hofman A, Homuth G, Humphries SE, Hyppönen E, Illig T, Jarvelin MR, Johansen B, Jousilahti P, Jula AM, Kaprio J, Kee F, Keinanen-Kiukaanniemi SM, Kooner JS, Kooperberg C, Kovacs P, Kraja AT, Kumari M, Kuulasmaa K, Kuusisto J, Lakka TA, Langenberg C, Le Marchand L, Lehtimäki T, Lyssenko V, Männistö S, Marette A, Matise TC, McKenzie CA, McKnight B, Musk AW, Möhlenkamp S, Morris AD, Nelis M, Ohlsson C, Oldehinkel AJ, Ong KK, Palmer LJ, Penninx BW, Peters A, Pramstaller PP, Raitakari OT, Rankinen T, Rao DC, Rice TK, Ridker PM, Ritchie MD, Rudan I, Salomaa V, Samani NJ, Saramies J, Sarzynski MA, Schwarz PE, Shuldiner AR, Staessen JA, Steinthorsdottir V, Stolk RP, Strauch K, Tönjes A, Tremblay A, Tremoli E, Vohl MC, Völker U, Vollenweider P, Wilson JF, Witteman JC, Adair LS, Bochud M, Boehm BO, Bornstein SR, Bouchard C, Cauchi S, Caulfield MJ, Chambers JC, Chasman DI, Cooper RS, Dedoussis G, Ferrucci L, Froguel P, Grabe HJ, Hamsten A, Hui J, Hveem K, Jöckel KH, Kivimaki M, Kuh D, Laakso M, Liu Y, März W, Munroe PB, Njølstad I, Oostra BA, Palmer CN, Pedersen NL, Perola M, Pérusse L, Peters U, Power C, Quertermous T, Rauramaa R, Rivadeneira F, Saaristo TE, Saleheen D, Sinisalo J, Slagboom PE, Snieder H, Spector TD, Stefansson K, Stumvoll M, Tuomilehto J, Uitterlinden AG, Uusitupa M, van der Harst P, Veronesi G, Walker M, Wareham NJ, Watkins H, Wichmann HE, Abecasis GR, Assimes TL, Berndt SI, Boehnke M, Borecki IB, Deloukas P, Franke L, Frayling TM, Groop LC, Hunter DJ, Kaplan RC, O'Connell JR, Qi L, Schlessinger D, Strachan DP, Thorsteinsdottir U, van Duijn CM, Willer CJ, Visscher PM, Yang J, Hirschhorn JN, Zillikens MC, McCarthy MI, Speliotes EK, North KE, Fox CS, Barroso I, Franks PW, Ingelsson E, Heid IM, Loos RJ, Cupples LA, Morris AP, Lindgren CM, Mohlke KL.Nature. 2015 Feb 12;518(7538):187-196. doi: 10.1038/nature14132.PMID: 25673412 Free PMC article. To increase our understanding of the genetic basis of body fat distribution and its molecular links to cardiometabolic traits, here we conduct genome-wide association meta-analyses of traits related to waist and hip circumferences in up to 224,459 individuals. ...   4 Cite   Share   Sex-dimorphic genetic effects and novel loci for fasting glucose and insulin variability. Lagou V, Mägi R, Hottenga JJ, Grallert H, Perry JRB, Bouatia-Naji N, Marullo L, Rybin D, Jansen R, Min JL, Dimas AS, Ulrich A, Zudina L, Gådin JR, Jiang L, Faggian A, Bonnefond A, Fadista J, Stathopoulou MG, Isaacs A, Willems SM, Navarro P, Tanaka T, Jackson AU, Montasser ME, O'Connell JR, Bielak LF, Webster RJ, Saxena R, Stafford JM, Pourcain BS, Timpson NJ, Salo P, Shin SY, Amin N, Smith AV, Li G, Verweij N, Goel A, Ford I, Johnson PCD, Johnson T, Kapur K, Thorleifsson G, Strawbridge RJ, Rasmussen-Torvik LJ, Esko T, Mihailov E, Fall T, Fraser RM, Mahajan A, Kanoni S, Giedraitis V, Kleber ME, Silbernagel G, Meyer J, Müller-Nurasyid M, Ganna A, Sarin AP, Yengo L, Shungin D, Luan J, Horikoshi M, An P, Sanna S, Boettcher Y, Rayner NW, Nolte IM, Zemunik T, Iperen EV, Kovacs P, Hastie ND, Wild SH, McLachlan S, Campbell S, Polasek O, Carlson O, Egan J, Kiess W, Willemsen G, Kuusisto J, Laakso M, Dimitriou M, Hicks AA, Rauramaa R, Bandinelli S, Thorand B, Liu Y, Miljkovic I, Lind L, Doney A, Perola M, Hingorani A, Kivimaki M, Kumari M, Bennett AJ, Groves CJ, Herder C, Koistinen HA, Kinnunen L, Faire U, Bakker SJL, Uusitupa M, Palmer CNA, Jukema JW, Sattar N, Pouta A, Snieder H, Boerwinkle E, Pankow JS, Magnusson PK, Krus U, Scapoli C, de Geus EJCN, Blüher M, Wolffenbuttel BHR, Province MA, Abecasis GR, Meigs JB, Hovingh GK, Lindström J, Wilson JF, Wright AF, Dedoussis GV, Bornstein SR, Schwarz PEH, Tönjes A, Winkelmann BR, Boehm BO, März W, Metspalu A, Price JF, Deloukas P, Körner A, Lakka TA, Keinanen-Kiukaanniemi SM, Saaristo TE, Bergman RN, Tuomilehto J, Wareham NJ, Langenberg C, Männistö S, Franks PW, Hayward C, Vitart V, Kaprio J, Visvikis-Siest S, Balkau B, Altshuler D, Rudan I, Stumvoll M, Campbell H, van Duijn CM, Gieger C, Illig T, Ferrucci L, Pedersen NL, Pramstaller PP, Boehnke M, Frayling TM, Shuldiner AR, Peyser PA, Kardia SLR, Palmer LJ, Penninx BW, Meneton P, Harris TB, Navis G, Harst PV, Smith GD, Forouhi NG, Loos RJF, Salomaa V, Soranzo N, Boomsma DI, Groop L, Tuomi T, Hofman A, Munroe PB, Gudnason V, Siscovick DS, Watkins H, Lecoeur C, Vollenweider P, Franco-Cereceda A, Eriksson P, Jarvelin MR, Stefansson K, Hamsten A, Nicholson G, Karpe F, Dermitzakis ET, Lindgren CM, McCarthy MI, Froguel P, Kaakinen MA, Lyssenko V, Watanabe RM, Ingelsson E, Florez JC, Dupuis J, Barroso I, Morris AP, Prokopenko I; Meta-Analyses of Glucose and Insulin-related traits Consortium (MAGIC).Nat Commun. 2021 Jan 5;12(1):24. doi: 10.1038/s41467-020-19366-9.PMID: 33402679 Free PMC article.     5 Cite   Share   Genome-wide associations for birth weight and correlations with adult disease. Horikoshi M, Beaumont RN, Day FR, Warrington NM, Kooijman MN, Fernandez-Tajes J, Feenstra B, van Zuydam NR, Gaulton KJ, Grarup N, Bradfield JP, Strachan DP, Li-Gao R, Ahluwalia TS, Kreiner E, Rueedi R, Lyytikäinen LP, Cousminer DL, Wu Y, Thiering E, Wang CA, Have CT, Hottenga JJ, Vilor-Tejedor N, Joshi PK, Boh ETH, Ntalla I, Pitkänen N, Mahajan A, van Leeuwen EM, Joro R, Lagou V, Nodzenski M, Diver LA, Zondervan KT, Bustamante M, Marques-Vidal P, Mercader JM, Bennett AJ, Rahmioglu N, Nyholt DR, Ma RCW, Tam CHT, Tam WH; CHARGE Consortium Hematology Working Group, Ganesh SK, van Rooij FJ, Jones SE, Loh PR, Ruth KS, Tuke MA, Tyrrell J, Wood AR, Yaghootkar H, Scholtens DM, Paternoster L, Prokopenko I, Kovacs P, Atalay M, Willems SM, Panoutsopoulou K, Wang X, Carstensen L, Geller F, Schraut KE, Murcia M, van Beijsterveldt CE, Willemsen G, Appel EVR, Fonvig CE, Trier C, Tiesler CM, Standl M, Kutalik Z, Bonas-Guarch S, Hougaard DM, Sánchez F, Torrents D, Waage J, Hollegaard MV, de Haan HG, Rosendaal FR, Medina-Gomez C, Ring SM, Hemani G, McMahon G, Robertson NR, Groves CJ, Langenberg C, Luan J, Scott RA, Zhao JH, Mentch FD, MacKenzie SM, Reynolds RM; Early Growth Genetics (EGG) Consortium, Lowe WL Jr, Tönjes A, Stumvoll M, Lindi V, Lakka TA, van Duijn CM, Kiess W, Körner A, Sørensen TI, Niinikoski H, Pahkala K, Raitakari OT, Zeggini E, Dedoussis GV, Teo YY, Saw SM, Melbye M, Campbell H, Wilson JF, Vrijheid M, de Geus EJ, Boomsma DI, Kadarmideen HN, Holm JC, Hansen T, Sebert S, Hattersley AT, Beilin LJ, Newnham JP, Pennell CE, Heinrich J, Adair LS, Borja JB, Mohlke KL, Eriksson JG, Widén EE, Kähönen M, Viikari JS, Lehtimäki T, Vollenweider P, Bønnelykke K, Bisgaard H, Mook-Kanamori DO, Hofman A, Rivadeneira F, Uitterlinden AG, Pisinger C, Pedersen O, Power C, Hyppönen E, Wareham NJ, Hakonarson H, Davies E, Walker BR, Jaddoe VW, Jarvelin MR, Grant SF, Vaag AA, Lawlor DA, Frayling TM, Davey Smith G, Morris AP, Ong KK, Felix JF, Timpson NJ, Perry JR, Evans DM, McCarthy MI, Freathy RM.Nature. 2016 Oct 13;538(7624):248-252. doi: 10.1038/nature19806. Epub 2016 Sep 28.PMID: 27680694 Free PMC article. Birth weight (BW) has been shown to be influenced by both fetal and maternal factors and in observational studies is reproducibly associated with future risk of adult metabolic diseases including type 2 diabetes (T2D) and cardiovascular disease. ...In addition, using large …   6 Cite   Share   Physical activity and cardiovascular risk factors in Spanish children aged 11-13 years. Cordova A, Villa G, Sureda A, Rodriguez-Marroyo JA, Sánchez-Collado MP.Rev Esp Cardiol (Engl Ed). 2012 Jul;65(7):620-6. doi: 10.1016/j.recesp.2012.01.026. Epub 2012 May 24.PMID: 22633280 English, Spanish. Children were allowed to join one of the following groups: a) sedentary group (2h/week of physical education at school); b) active group (2h/week of physical education at school plus 3h/week extra physical activity), and c) sports group (2h/week …   7 Cite   Share   Cluster Analysis of Physical Activity Patterns, and Relationship with Sedentary Behavior and Healthy Lifestyles in Prepubertal Children: Genobox Cohort. Leis R, Jurado-Castro JM, Llorente-Cantarero FJ, Anguita-Ruiz A, Iris-Rupérez A, Bedoya-Carpente JJ, Vázquez-Cobela R, Aguilera CM, Bueno G, Gil-Campos M.Nutrients. 2020 May 1;12(5):1288. doi: 10.3390/nu12051288.PMID: 32370020 Free PMC article. Cluster analysis was performed to establish different groups based on physical activity levels. A total of 489 children were finally selected. ...The choice to practice an extracurricular sport could be an influencing factor to increase exercise … Item in Clipboard 8 Cite   Share   Associations of MC4RLEP, and LEPR Polymorphisms with Obesity-Related Parameters in Childhood and Adulthood. Raskiliene A, Smalinskiene A, Kriaucioniene V, Lesauskaite V, Petkeviciene J.Genes (Basel). 2021 Jun 21;12(6):949. doi: 10.3390/genes12060949.PMID: 34205732 Free PMC article. In childhood, only skinfold thicknesses were associated with gene variants being the lowest in children with MC4R TT genotype and LEP AG genotype. In adulthood, odds of obesity and metabolic syndrome was higher in MC4R CT/CC genotype than TT genotype carriers (OR 1. … Item in Clipboard 9 Cite   Share   Mechanisms for Sex Differences in Energy Homeostasis. Wang C, Xu Y.J Mol Endocrinol. 2019 Feb;62(2):R129-R143. doi: 10.1530/JME-18-0165. Epub 2019 Feb 1.PMID: 31130779 Free PMC article. Review. Better understanding of the underlying mechanisms for sexual dimorphism in energy balance may facilitate development of gender-specific therapies for human diseases, e.g. obesity. Multiple organs, including the brain, liver, fat and muscle, play important roles in the regu …   10 Cite   Share   GHS-R suppression in adipose tissues protects against obesity and insulin resistance by regulating adipose angiogenesis and fibrosis. Lee JH, Fang C, Li X, Wu CS, Noh JY, Ye X, Chapkin RS, Sun K, Sun Y.Int J Obes (Lond). 2021 Jul;45(7):1565-1575. doi: 10.1038/s41366-021-00820-

Author Response

Dear Ms. Fia Cheng

Section Managing Editor,

Thank you for providing us with the opportunity to submit a revised version of our editorial entitled The role of molecular and hormonal factors in obesity and the effects of physical activity in childrento the International Journal of Molecular Sciences in the Special Issue “Frontiers in Obesity”.

We would like to thank the reviewers for their thoughtful comments and suggestions regarding our manuscript. All comments received from reviewers have been incorporated into the revised manuscript. Below is an itemized point-by-point response to the comments from the reviewers in response to the changes made to the original document (highlighted in yellow).

COMMENTS FROM REVIEWER 2

Comment #1

This is a good paper. Only add these references more and try to explain them at discussion topics

-Genetic studies of body mass index yield new insights for obesity biology. Locke AE, Kahali B, Berndt SI, Justice AE, Pers TH, Day FR, Powell C, Vedantam S, Buchkovich ML, Yang J, et al.Nature. 2015 Feb 12;518(7538):197-206. doi: 10.1038/nature14177.

-FTO Obesity Variant Circuitry and Adipocyte Browning in Humans. Claussnitzer M, Dankel SN, Kim KH, Quon G, Meuleman W, Haugen C, Glunk V, Sousa IS, Beaudry JL, Puviindran V, Abdennur NA, Liu J, Svensson PA, Hsu YH, Drucker DJ, Mellgren G, Hui CC, Hauner H, Kellis M.N Engl J Med. 2015 Sep 3;373(10):895-907. doi: 10.1056/NEJMoa1502214.

-New genetic loci link adipose and insulin biology to body fat distribution. Shungin D, Winkler TW, Croteau-Chonka DC, Ferreira T, Locke AE, Mägi R, Strawbridge RJ, Pers TH, Fischer K, Justice AE, et al. Nature. 2015 Feb 12;518(7538):187-196. doi: 10.1038/nature14132.

-Sex-dimorphic genetic effects and novel loci for fasting glucose and insulin variability. Lagou V, Mägi R, Hottenga JJ, Grallert H, Perry JRB, Bouatia-Naji N, Marullo L, Rybin D, Jansen R, Min JL, et al. Nat Commun. 2021 Jan 5;12(1):24. doi: 10.1038/s41467-020-19366-9.

-Genome-wide associations for birth weight and correlations with adult disease. Horikoshi M, Beaumont RN, Day FR, Warrington NM, Kooijman MN, Fernandez-Tajes J, Feenstra B, van Zuydam NR, Gaulton KJ, Grarup N, et al. Nature. 2016 Oct 13;538(7624):248-252. doi: 10.1038/nature19806.

-Physical activity and cardiovascular risk factors in Spanish children aged 11-13 years. Cordova A, Villa G, Sureda A, Rodriguez-Marroyo JA, Sánchez-Collado MP.Rev Esp Cardiol (Engl Ed). 2012 Jul;65(7):620-6. doi: 10.1016/j.recesp.2012.01.026.

-Cluster Analysis of Physical Activity Patterns, and Relationship with Sedentary Behavior and Healthy Lifestyles in Prepubertal Children: Genobox Cohort.Leis R, Jurado-Castro JM, Llorente-Cantarero FJ, Anguita-Ruiz A, Iris-Rupérez A, Bedoya-Carpente JJ, Vázquez-Cobela R, Aguilera CM, Bueno G, Gil-Campos M.Nutrients. 2020 May 1;12(5):1288. doi: 10.3390/nu12051288.

-Associations of MC4RLEP, and LEPR Polymorphisms with Obesity-Related Parameters in Childhood and Adulthood. Raskiliene A, Smalinskiene A, Kriaucioniene V, Lesauskaite V, Petkeviciene J.Genes (Basel). 2021 Jun 21;12(6):949. doi: 10.3390/genes12060949.

-Mechanisms for Sex Differences in Energy Homeostasis. Wang C, Xu Y.J Mol Endocrinol. 2019 Feb;62(2):R129-R143. doi: 10.1530/JME-18-0165.

-GHS-R suppression in adipose tissues protects against obesity and insulin resistance by regulating adipose angiogenesis and fibrosis. Lee JH, Fang C, Li X, Wu CS, Noh JY, Ye X, Chapkin RS, Sun K, Sun Y. Int J Obes (Lond). 2021 Jul;45(7):1565-1575. doi: 10.1038/s41366-021-00820-

Response: I would like to thank the reviewer for his/her kind comments regarding our manuscript and revisions. As a result of the reviewer's comments, the used citations (9 of 10) have been added to the main document and highlighted in yellow. The last reference (Int J Obes 2021 Jul;45(7):1565-1575) has been omitted since ghrelin is not discussed in the text.
